



# Linear Optimal Runoff Aggregate (LORA): A global gridded synthesis runoff product

Sanaa Hobeichi[1,2], Gab Abramowitz[1,3], Jason Evans[1,3] and Hylke E. Beck[4]

[1] Climate Change Research Centre, University of New South Wales, Sydney, NSW 2052, Australia
5    [2] ARC Centre of Excellence for Climate System Science, University of New South Wales, Sydney, NSW 2052, Australia
[3] ARC Centre of Excellence for Climate Extremes, University of New South Wales, Sydney, NSW 2052, Australia
[4] Department of Civil and Environmental Engineering, Princeton University, Princeton, NJ 08544, USA

*Correspondence to*: Sanaa Hobeichi (s.hobeichi@student.unsw.edu.au)

10    **Abstract**

No synthesized global gridded runoff product, derived from multiple sources, is available despite such a product being useful to meet the needs of many global water initiatives. We apply an optimal weighting approach to merge runoff estimates from hydrological models constrained with observational streamflow records. The weighting method is based on the ability of the models to match observed streamflow data while accounting for error covariance between the participating products. To 15   address the lack of observed streamflow for many regions, a dissimilarity method was applied to transfer the weights of the participating products to the ungauged basins from the closest gauged basins using dissimilarity between basins in physiographic and climatic characteristics as a proxy for distance. We perform out-of-sample tests to examine the success of the dissimilarity approach and we confirm that the weighted product performs better than its 11 constituents products in a range of metrics. Our resulting synthesized global gridded runoff product is available at monthly time scales, and includes 20   time variant uncertainty, for the period 1980 – 2012 on a 0.5º grid. The synthesized global gridded runoff product broadly agrees with published runoff estimates at many river basins, and represents well the seasonal runoff cycle for most of the globe. The new product, called Linear Optimal Runoff Aggregate (LORA), is a valuable synthesis of existing runoff products and will be freely available for download on geonetwork.nci.org.au.

**Introduction**

25   Runoff is the horizontal flow of water on land or through soil before it reaches a stream, river, lake, reservoir or other channels. It has been widely used as a metric for droughts (Shukla and Wood, 2008; van Huijgevoort et al., 2013; Bai et al., 2014; Ling et al., 2016) and to understand the effects of climate change on the hydrological cycle (Ukkola et al., 2016; Zhai and Tao, 2017). Accurate estimates of runoff are critical to inform climate change adaptation strategies, to guide appropriate



water management in agriculture (Nyamadzawo et al., 2012), and to enable the assessment of the impact of anthropogenic activities on ecosystems (Vörösmarty et al., 2010), yet direct measurement of runoff at large scales is simply not possible.

While runoff observations do not exist, direct streamflow or river discharge observations - basin integrated runoff - have been archived in many databases. The most comprehensive international streamflow database is the Global Runoff Data
Base (GRDB; www.bafg.de), which consists of daily and monthly quality-controlled streamflow records from more than 9500 gauges across the globe. Geospatial Attributes of Gages for Evaluating Streamflow version II (GAGES-II) represents another noteworthy streamflow database, consisting of daily quality-controlled streamflow data from over 9000 US gauges. Hydrological and land surface models are capable of producing gridded runoff estimates for any region across the globe (Sood and Smakhtin, 2015; Liu et al., 2016; Kauffeldt et al., 2016). However, these runoff estimates suffer from
uncertainties due to shortcomings in the model structure and parameterization and the meteorological forcing data (Beven, 1989; Beck, 2017a). There are various ways to use streamflow observations for improving the runoff outputs from these models. The conventional approach consists of model parameter calibration using locally observed streamflow data (see review by Pechlivanidis et al., 2011). Another widely used method is through regionalization; that is, the transfer of knowledge (e.g., calibrated parameters) from gauged basins to ungauged basins (see review by Beck et al., 2016). In
contrast, several other studies attempted to correct the runoff outputs directly rather than the model parameters, for example by bias-correcting model runoff outputs based on streamflow observations ( see review by Ye et al., 2014), or by combining or weighting ensembles of model outputs to obtain improved runoff estimates (e.g., Aires, 2014). There are, however, relatively few continental- and global-scale efforts to improve model estimates using observed streamflow.

A broad array of gridded model-based runoff estimates are freely available, including ECMWF's Interim reanalysis (ERA-
Interim ; Dee et al., 2011), NASA's Modern Era Retrospective-analysis for Research and Applications ( MERRA) Land (Reichle et al., 2011), the Climate Forecast System Reanalysis (CFSR; Tomy and Sumam, 2016), the second global soil wetness project (GSWP2; Dirmeyer et al., 2006), the Water Model Intercomparison Project (WaterMIP; Haddeland et al., 2011), and the Global Land Data Assimilation System (GLDAS; Rodell et al., 2004). Recently, the eartH2Observe project has made available two ensembles (tier-1 and -2) of state-of-the-art global hydrological and land surface model outputs
(http://www.earth2observe.eu/; Beck et al., 2017a; and Schellekens et al., 2017). Although these model simulations represent the only time varying gridded estimates of runoff at the global scale, they are subject to considerable uncertainties, resulting in large differences in runoff simulated by the models. Many studies have therefore evaluated and compared the gridded runoff models (see overview in Table 1 of Beck et al., 2017a).

Despite the demonstrated improved predictive capability of multi-model ensemble approaches (Sahoo et al., 2011; Pan et al.,
2012; Bishop and Abramowitz, 2013; Mueller et al., 2013; Munier et al., 2014; Aires, 2014; Rodell et al., 2015; Jiménez et al., 2017; Hobeichi et al., 2018; Zhang et al., 2018), very little has been done to utilise this range of model simulations toward improved runoff estimates. This paper implements the weighting and rescaling method introduced in Bishop and



Abramowitz (2013) and Abramowitz and Bishop (2015) to derive a monthly 0.5° global synthesis runoff product. Briefly summarized, we use a bias correction and weighting approach to merge 11 state-of-the-art gridded runoff products from the eartH2Observe project, constrained by observed streamflow from a variety of sources. This approach also provides us with corresponding uncertainty estimates that are better constrained than the simple range of modelled values. For ungauged

regions we employ a dissimilarity method to transfer the product weights to the ungauged basins from the closest basins using dissimilarity between basins as a proxy for distance. Such a synthesis product is in line with the multi-source strategy of Global Energy and Water EXchanges (GEWEX; Morel, 2001) and NASA's Making Earth Science Data Records for Use in Research Environments (MEaSUREs; Earthdata, 2017) initiatives and is particularly useful for studies that aim to close the water budget at the grid scale.

Sections 2.1 describes the observed streamflow data. Section 2.2 presents the participating datasets used to derive the weighted runoff product. Section 2.3 details the weighting method implemented in the gauged basins, while Section 2.4 focuses on the ungauged basins. Section 2.5 examines the approach used to derive the global runoff product. We then present and discuss our results in Section 3 and 4 before concluding.

**Data and Methods**

**2.1 Observed streamflow data**

We used observed streamflow from the following four sources: (i) the US Geological Survey (USGS) Geospatial Attributes of Gages for Evaluating Streamflow (GAGES)-II database (Falcone et al., 2010); (ii) the Global Runoff Data Base (GRDB; http://www.bafg.de/GRDC/); (iii) the Australian Peel et al. (2000) database; and (iv) the global Dai (2016) database. We discarded duplicates and from the remaining set of stations discarded those satisfying at least one of the following criteria: (i)

basin area <8000 km$^2$ (fewer than three 0.5° grid cells); (ii) record length <5 y in the period 1980–2012 (not necessarily consecutive); and (iii) low observed streamflow (i.e. around 0) that does not represent the total runoff across the basins due to significant anthropogenic activities. A river basin was identified with significant anthropogenic activities if it has > 20% irrigated area using the Global Map of Irrigation Areas (GMIA-Version 4.0.2; Siebert et al., 2007) or has > 20% classified as "Artificial surfaces and associated areas" according to the Global Land Cover Map (GlobCover-Version 2.3; Bontemps et

al., 2011). In total 596 stations (of which 20 are nested in the basins of other stations) were found to be suitable for the analysis (Fig. 1).

**2.2 Simulated runoff data**

To derive the global monthly 0.5° synthesis runoff product, we used 11 total runoff outputs (from eight different models) and

seven streamflow outputs (from six different models) produced as part of tiers 1 and 2 of the eartH2Observe project (available via ftp://wci.earth2observe.eu/). The models and their available variables are presented in Table 1. For tier 1 of





eartH2Observe, the models were forced with the WATCH Forcing Data ERA-Interim (WFDEI) meteorological dataset (Weedon et al., 2014) corrected using the Climatic Research Unit Timeseries dataset (CRU-TS3.1; Harris et al., 2014). For tier 2, the models were forced using the Multi-Source Weighted-Ensemble Precipitation (MSWEP) dataset (Beck et al., 2017b). The runoff and streamflow values are provided in kg m$^{-2}$s$^{-1}$ and m$^3$ s$^{-1}$, respectively. For consistency, the runoff

outputs with resolution <0.5° were resampled to 0.5° using bilinear interpolation. In some cases, the river network employed by the model did not correspond with the stream gauge location, in which case we manually selected the grid cell that provided the best match with the observed streamflow.

The runoff outputs were only used if no streamflow output was available and only in basins smaller than 100,000 km². To make the runoff data consistent with the streamflow data, we integrated the runoff over the basin areas (termed Ragg, units

m$^3$ s$^{-1}$). Thus, for basins smaller than 100,000 km² the synthesis product was derived from 11 model outputs, whereas for basins larger than 100,000 km² the synthesis product was derived from seven outputs.

**2.3 Implementing the weighting approach at the gauged basins**

At each gauged basin, we built a linear combination $\mu_q$ of the participating modelled streamflow datasets $x$ (i.e. Ragg in small basins and modelled streamflow, $q$, in large basins) that minimized the mean square difference with the observed

streamflow $Q$ at that basin such that: $\mu_q^j = \sum_{k=1}^{K} w_k(x_k^j - b_k)$ where $j \in [1, J]$ are the time steps and $k \in [1, K]$ represent the participating models, $x_k^j$ (i.e., Ragg$_k^j$ in small basins and $q_k^j$ in large basins) is the value of the participating dataset in m$^3$ s$^{-1}$ at the $j$th time step of the $k$th participating model, the bias term $b_k$ is the mean error of $x_k$ in m$^3$ s$^{-1}$. The set of weights $w_k$ provides an analytical solution to the minimization of $\sum_{j=1}^{J}(\mu_q^j - Q^j)^2$, where $Q^j$ is the observed streamflow at the $j$th time step (for derivation see Bishop and Abramowitz (2013)).

We then derived the weighted runoff dataset by applying the computed weights on the bias corrected runoff estimates of the participating models. The weighted runoff dataset is expressed as:

$$\mu_r^j = \sum_{k=1}^{K} w_k(r_k^j - b'_k)$$

Where $r_k^j$ is the value of runoff estimate in kg m$^{-2}$s$^{-1}$ of the $k$th participating model at the $j$th time step and b′$_k$ is its runoff bias in kg m$^{-2}$s$^{-1}$.

To calculate the runoff bias b′$_k$, we assumed that for each model $k$ and at each time $j$ the bias ratio of a model (defined as the ratio of the model error to the simulated magnitude) is the same for streamflow and runoff estimates Eq. (1). In small basins, the bias ratio of modeled streamflow was calculated by using Ragg$_k^j$ instead of the modeled streamflow $q_k^j$ Eq. (2).

$$[\frac{q_k^j - Q^j}{q_k^j} = \frac{b'_k}{r_k^j}]_{basin} \qquad (1)$$



$$\left[\frac{\mathrm{Ragg}_k^j - Q^j}{\mathrm{Ragg}_k^j} = \frac{b\prime_k}{r_k^j}\right]_{\mathrm{basin}} \qquad (2)$$

To avoid over-fitting when applying the weighting approach, we limited the number of participating models so that the ratio of number of records to number of models does not fall below 10. As a result of this, when required, we discarded the models that had the highest bias (i.e. left terms in Eq (1, 2)) until the threshold was met. The weighting and the bias

correction occasionally resulted in negative runoff values, we replaced any negative values with zero.

We implemented the ensemble dependence transformation process detailed in Bishop and Abramowitz (2013) to compute the gridded time-variant uncertainty associated with the derived runoff estimates, following the same approach as in Hobeichi et al. (2018). For any particular gauged basin, we first calculated the spatial aggregate of our weighted runoff estimate, $R_{\mathrm{agg}_\mu}$, then quantified $s_r^2$, the error variance of $R_{\mathrm{agg}_\mu}$ with respect to the observed streamflow $Q$ over time and

space. We then transformed the constituent modelled estimates so that their variance about $R_{\mathrm{agg}_\mu}$ at a given time step $\sigma_r^{2j}$, averaged over all time steps where we have available streamflow data for the current basin, is equal to $s_r^2$. This transformed ensemble provides us with uncertainty estimates that (a) are varying in time and space, and (b) accurately reflects our ability to reproduce the observed streamflow. It provides a much more defensible uncertainty estimate than simply calculating the standard deviation of the involved products. We then used $\sqrt{\sigma_r^{2j}}$ as the spatially and temporally varying estimate of

uncertainty standard deviation, which we will refer to below simply as 'uncertainty'. For more details about how this technique was implemented we refer readers to Hobeichi et al. (2018).

### 2.4 Deriving runoff estimates at the ungauged river basins

Implementing the weighting approach requires observed streamflow to constrain the weighting, which we do not have at ungauged river basins (defined in section 2.1). To address this, we used the modelled and observed streamflow from the

three most similar gauged river basins, based on pre-defined physical and climatic characteristics, to derive model weights at each ungauged basin. The selected gauged river basins served as donor basins to the ungauged receptor basins. We then implemented the weighting technique on the ensemble of 11 (in small basins) or eight (in large basins) model outputs by matching Ragg calculated across the selected donor basins with the observed streamflow. Finally, we transferred the weights and bias ratios computed at the donor basins to the receptor basin and subsequently computed the associated uncertainty

values.

Most of the gauged river basins were classified as donor basins. Some, however, were excluded from being donors where we found (based on Ragg or modeled streamflow time series and metric values) that none of the models was able to simulate the streamflow dynamics. These basins are mainly located in areas of natural lakes, in mountainous areas covered with snow, or in wet regions with intense rainfall. We therefore (subjectively) decided that those excluded basins should be assigned to a

"non–donor and non–receptor" category.





We applied the method presented in Beck et al. (2016) to calculate a similarity index $S$ between a donor basin $a$ and a receptor basin $b$ expressed as:

$$S_{a,b} = \sum_{p=1}^{7} \frac{|Z_{p,a} - Z_{p,b}|}{IQR_p} \qquad (3)$$

Where $p$ denotes the climatic and physiographic characteristics as in Table 4 of Beck et al. (2016). This includes aridity index, fractions of forest and snow cover, soil clay content, surface slope, and annual averages of precipitation and potential evaporation. $Z_{p,a}$ and $Z_{p,b}$ are the values of the characteristic $p$ at donor and receptor basins, respectively. $IQR_p$ is the interquartile range of characteristic $p$ calculated over the land surface, excluding deserts (defined by an aridity index > 5, see Table 4 of Beck et al. (2016)) and areas with permanent ice (defined by climate zones Tundra, Subarctic and Ice cap using a simplified climate zones map created by the Esri Education Team for ArcGIS online (World Climate Zones – Simplified; Esri Education Team, 2014)). From Eq. 3 it follows that the most similar donor $a$ to a receptor $b$ is the one that has the lowest index value with basin $b$. We applied this approach to identify the 3 most similar donors for every receptor basin.

In very large basins, physiographic and climatic heterogeneity can result in misleading basin-mean averages. We therefore excluded highly heterogeneous basins from the list of donors and classified them as 'non-donor and non-receptor' basins, and also broke up large heterogeneous receptor basins by climate groups into smaller basin zones and then treated them as separate basins to effectively receive sets of weights and bias ratios from the donor basins to the separate parts. Here we defined large heterogeneous basins as basins with areas greater than 1,000,000 km$^2$ and covering climate zones that belong to at least two groups of 1) Tropical Wet, 2) Humid continental, Humid subtropical, Mediterranean and Marine, 3) Tropical Dry, Semi–arid and Arid, 4) Tundra, Subarctic and Ice cap and 5) Highlands. Climate classification is based on the simplified climate zones map (World Climate Zones te zones map; Esri Education Team, 2014) defined above. Figure 2 shows the spatial coverage of the donor basins, receptor basins and non-donor and non-receptor basins, and Fig. 3 summarizes the steps carried out to derive the weighted runoff product for the global land.

## 2.5 Out-of-sample testing

To test that this approach is producing a runoff estimate at receptor basins (using transferred weights from the most similar gauged basins) that is better than any of the individual models, we performed an out-of-sample test. In this test, we selected a gauged basin and treated it as a receptor basin, constructing model weights by using the three most similar donor basins. We could then compare: (a) observed streamflow; (b) the in-sample weighted product (WP$_{in}$) derived by using observed streamflow for this basin to weight models; (c) an out-of-sample weighted product (WP$_{out}$) derived by constructing the weighting at the three most similar basins, and; (d) the individual model estimates at each basin. We calculated four metrics of performance for WP$_{in}$, WP$_{out}$ and each of the 11 datasets: Mean Square Error  MSE=mean(Ragg – observed streamflow)$^2$; Mean Bias=mean| Ragg – observed streamflow |; Correlation COR=corr(observed streamflow, Ragg) and Standard Deviation (SD) difference= $\sigma_{Ragg} - \sigma_{observed\ streamflow}$. We repeated the out-of-sample test for all the gauged basins (donor basins and non-donor and non-receptor basins).



We displayed the results of the out-sample-test by showing the percentage performance improvement of $WP_{out}$ compared to $WP_{in}$ and each individual model, yielding 12 different values of performance improvement. If the approach is succeeding, we expect that both $WP_{out}$ and $WP_{in}$ will perform better than any of the models used in this study, and also $WP_{in}$ should be in better agreement with the observed streamflow when compared to $WP_{out}$.

We used box and whisker plots to show the results of performance improvement of $WP_{out}$ calculated relative to $WP_{in}$ and the 11 datasets across all the gauged basins. The lower and upper hinges of a boxplot represent the first ($Q_1$) and third ($Q_3$) quartiles respectively of the performance improvement results and the line inside the boxplot shows the median value. The extreme of the lower whisker represents the maximum of 1) min(dataset) and 2) ($Q_1$ - IQR), while the extreme of the upper whisker is the minimum of 1) max(dataset) and 2) ($Q_3$ + IQR)), where IQR represents the interquartile range (i.e. $Q_3$ - $Q_1$ ) of

the performance improvement results. A median line located above the 0 axis is an indication that the out of sample weighting offers an improvement in more than half of the basins.

## 3 Results

The results for the out-of-sample test are displayed in the box and whisker plots presented in Fig. 4 (a - d).

The MSE and Mean bias plots in Fig. 4 (a and d) indicate that across almost all the gauged basins $WP_{out}$ performs better than

each of the individual models. Similarly, the COR plot in Fig. 3 (c) shows that the out-of-sample weighting has in fact improved the correlation with observational data across almost all the gauged basins. The SD difference plot (Fig. 4 (b)) shows a significant improvement of $WP_{out}$ relative to the models, but the number of basins that benefit from this improvement decreased, perhaps because the variability of the individual members of the weighting ensemble is not necessarily temporally coincident at all the basins, resulting in decreased variability. The negative performance improvement

of $WP_{out}$ relative to $WP_{in}$ across all metrics (first boxplot, Fig. 4 (a-d)) indicates that the weighting performs better in-sample than out-of sample, which is to be expected. Critically though, the fact that the weighting delivers improvement over all models when the weights are transferred from similar basins indicate that the dissimilarity technique is succeeding and can be effectively used at the ungauged basins by feeding the weighting with data from the most similar basins with streamflow observations.

Based on the improvement that the weighting approach implemented in both gauged and ungauged basins offers over Ragg estimates computed for 11 individual model runoff estimates, in terms of MSE, SD difference, COR and Mean Bias against observed streamflow data, we now present details of the mosaic of the individual weighted runoff estimates derived across all the basins that we name LORA. At the gauged basins, the weighting was trained with the Ragg of the modelled runoff at the individual basins and constrained with the observed streamflow. At ungauged basins, the dissimilarity approach was first

implemented to find the three most similar basins, then the weighting was trained on the combined datasets from these three



basins. Subsequently, weights were transferred to the ungauged basins and applied to combine the runoff estimates at the individual basins.

The eight modelled runoff datasets listed in Table 1 as part of the tier1 ensemble were recently included in a global evaluation by Beck et al. (2017a). In their analysis, they computed a summary performance statistic that they termed OS by incorporating several long-term runoff behavioural signatures defined in Table 3 of Beck et al. (2017a) and found that the mean of runoff estimates from four models only (LISFLOOD, WaterGAP3, W3RA and HBV-SIMREG) performed the best in terms of $\overline{OS}$ (i.e. mean of OS over all the basins included in their study) relative to each individual modelled runoff estimates and the mean of all the modelled runoff estimates. In this study, we calculated the mean runoff from the four best products found by Beck et al. (2017a), that is (LISFLOOD, WaterGAP3, W3RA and HBV-SIMREG. Hereafter, we refer this as "Best4", and we calculated four statistics (RMSE, SD difference, COR and Mean bias defined here as mean(dataset-obs)) for Ragg computed from LORA, Best4 and each of the 11 runoff datasets across all the gauged basins. The boxplots in Fig. 5 (a-d) display the results.

The RMSE plot in Fig. 5(a) shows that LORA has the lowest RMSE values with the observed streamflow. All of the component models exhibit a similar performance in RMSE. Similarly, LORA has overall the least SD difference with observations (Fig. 5 b) across more than half of the basins. The Mean bias plot in Fig. 5(d) shows a non-significant positive bias in LORA relative to the observation at the majority of the basins. Best4, HBV-SIMREG, PCR-GLOBWB and particularly LISFLOOD exhibit a positive mean bias across most of the basins but with much higher bias magnitude compared to that of LORA. HTESSEL and SURFEX estimates from both tiers (i.e tier1 and tier2) together with JULES (tier2) and WGAP3 show negative and positive bias distributed evenly across the basins. LORA shows the highest temporal correlation with the observed streamflow at more than half of gauged basins (Fig. 5 (c)). The low RMSE and Mean bias values relative to the other estimates is partly due to the bias correction applied before the weighting. While all the performance metrics calculated here show that LORA outperforms Best4, these metrics do not allow us to assess how well LORA performs in terms of bias in the runoff timing, replicating the peaks or representing quick runoff, with the exception of the correlation metric. These aspects were studied in more detail in Beck et al. (2017a) and showed that Best4 performs well in these performance metrics.

All the models involved in deriving LORA with the exception of HBV-SIMREG were found in the study of (Beck et al., 2017a) to show early spring snowmelt peak and an overall significant underestimation of runoff in the snow-dominated basins. To see how well LORA performs at high latitudes, we examined the gauged basins located at higher latitudes (>60°) and we calculated two statistics – COR and mean bias – as in Fig. 5 (c-d) but this time for the snow-dominated basins only. We display the results in Fig. 6.

The temporal correlation plot in Fig. 6 (a) shows that LORA is in better agreement with observed streamflow at snow-dominated basins compared to the ensemble of all the gauged basins on the globe (Fig. 5 (c)) with an overall average improvement of 7%. Similarly, HBV-SIMREG shows an improved correlation with the observed streamflow at snow-




dominated basins with an average improvement of 14%, this agrees with the results reported by Beck et al. (2017a) who attributed the improved performance of HBV-SIMREG in snow-dominated regions to a snowfall gauge undercatch correction. The overall performance of Best4 and LISFLOOD do not change in terms of spatial correlation; on the contrary, all the remaining products show a degraded performance. Figure 6 (b) shows that LORA exhibits small biases across snow-dominated basins relative to participating models. Conversely, with the exception of LISFLOOD, all the tier1 products including Best4 show a negative mean bias across more than half of the snow-dominated basin, in particular HTESSEL, JULES, SURFEX and W3RA show a large negative bias at most of these basins. This agrees with the negative bias found in the study of Beck et al. (2017a) in all tier1 products except LISFLOOD. These results indicate that LORA is likely to slightly overestimate runoff in high latitudes whereas all tier1 products with the exception of LISFLOOD tend to underestimate runoff in these regions, and that this underestimation is larger for HTESSEL, JULES, SURFEX and W3RA. Tier2 products show both positive and negative bias across the basins. Their bias is of a lower magnitude than that found in tier1 products. That is probably because the forcing precipitation used to derive tier 2 outputs (i.e. MSWEP) has less biases than that used to derive tier1 estimates (i.e. WFDEI corrected using CRU-TS3.1). We also calculated the two metrics, SD difference and mean bias as in Fig. 5 (a and b), but we found no noticeable differences in the performance of any of the products relative to that found globally in Fig. 5 (a and b). The results displayed in Fig. 5 and Fig. 6 are discussed further below.

We calculated the seasonal relative uncertainty expressed as the ratio of average uncertainty to mean runoff (i.e. $\frac{\text{mean runoff uncertainty}}{\text{mean runoff}}$) for the period 1980 – 2012. This metric is intended to show some indication of the reliability of the derived runoff, with results displayed in Fig. 7. Regions in red show grid cells that satisfy $\frac{\text{mean runoff uncertainty}}{\text{mean runoff}} < 1$, while those shown in yellow are regions where the value of mean runoff uncertainty are larger than the value of the associated mean runoff itself. Regions in blue are grid cells that have a zero mean runoff and hence an undetermined relative uncertainty. The global maps in Fig. 7 show a consistent low reliability in Sahel, Indus basin, Parana, the semi-arid regions of Eastern Argentina, Doring basin in South Africa, red river sub-basin of the Mississippi, Burdekin and Fitzroy basins in North-East Australia and many regions of the Arab Peninsula. The areas at the higher latitudes in Asia and North America show high reliability during Jun-Jul-Aug and low reliability during the rest of the year. Parts of Madeiry sub-basin – a major sub-basin of the Amazon – show low reliability during June-Nov. The basins in Central America show high reliability in all seasons except in Mar-May while River basins in Somalia show low reliability during the austral summer and winter. River basins in the far east show low reliability in spring and autumn and a higher reliability in winter and summer.

Figure 8 displays the seasonal cycles of Ragg for LORA and Best4 and the observed streamflow over 11 major river basins. To generate this plot, we calculated the average Ragg for each month over the period of availability of observed streamflow. The shaded regions represent the range of uncertainty aggregates associated with the derived runoff. In the Amazon basin, LORA overestimates runoff in the wet season and underestimates it in the dry season, but the observed streamflow during



the dry season still lies within the error bounds of LORA. LORA shows good agreement with the observed cycle in the Mississippi. In the Niger and Murray-Darling basins, while LORA overestimates the observed streamflow, it shows a much better agreement compared to Best4 which strongly overestimates runoff. In the Parana basin, LORA underestimates the observed streamflow in all seasons except summer. In the subarctic basins, LORA shows different behavior within the

individual basins. In Pechora and Olenek, LORA represents well the seasonal cycle and the magnitude of runoff, whereas in the Amur, Lena and Yenisei, LORA shows an early shift of the runoff peak and an overall overestimation of runoff. In the Indigirka, LORA overestimates the spring peak, but the observed seasonal cycle lies within the error bounds.

Finally, we compared our mean annual runoff (mm/year) with those estimated by a well-known land surface hydrological model the Variable Infiltration Capacity (VIC; Liang et al., 1994) model in the study of Zhang et al. (2018) over comparable temporal and spatial scale for 16 large basins chosen from different climate zones on the globe. The mean annual runoff was computed over the period 1984 - 2010 instead of 1980 – 2012 to maximize the temporal agreement with the study of Zhang et al. (2018).

Table 2 shows that for some basins VIC and LORA agree well in estimating mean annual runoff (i.e. difference between LORA and at least one of VIC and VIC adjusted for budget closure <10%). This threshold is met in the Amazon, Columbia, Congo, Danube, Mackenzie and Mississippi. The basins that show a larger difference between VIC and LORA but show that VIC estimates lie within the uncertainty bounds of LORA (i.e. between LORA-uncertainty and LORA+uncertainty) include the Indigirka, Olenek, Parana, Pechora, Yenisei and Yukon. Large discrepancies between VIC and LORA are found in Lena

and the Murray-Darling.

## 4 Discussion

The results of the out-of-sample test suggest that deriving runoff estimates in an ungauged basin by training the weighting with streamflow data from similar basins - in terms of climatic and physiographic characteristics - is successful. While the

runoff product derived by using weights from external basins outperforms the runoff estimates from the individual models, the weighted runoff derived in-sample offers overall even more capable runoff estimates.

It follows from Fig. 2 and Fig. 7 that the runoff values computed over dry climates tend to be less reliable than those in other regimes. This is perhaps due to the biases in the WFDEI precipitation forcing that intensify in the arid and semi-arid regions and propagate in the simulated runoff (Beck et al., 2017a). Also, due the lower density of gauged basins in the arid and semi-

arid climates compared to other regimes, receptor basins are dominant over dry climates, which reduces the skill of the weighting to produce good runoff estimates. This is also in line with our conclusions from Fig. 3 that the weighting provides more reliable results in the gauged basins.





All the tier1 model outputs involved in this study with the exception of HBV-SIMREG were found by Beck et al. (2017a) to show early spring snowmelt in the snow-dominated basins. Both the Yenisei and the Lena are large basins (2.6 and 2.4 million km², respectively), and hence – as noted in Sect. 2.2 – only models that had estimates of both streamflow and runoff were used to derive LORA at these basins, and therefore HBV-SIMREG – whose inclusion would have improved the

weighting - was excluded. Beck et al. (2017a) also found that LISFLOOD has the best square root-transformed mean annual runoff among the tier1 datasets and perfoms well in terms of temporal correlation in all climates, this agrees with the high temporal correlation of LISFLOOD seen in Fig. 5 (c) and Fig. 6 (a), and also explains the highest weights attributed to LISFLOOD in the majority of snow-dominated basins. Because of this, and because LISFLOOD tends to overestimate runoff across half of the snow-dominated basins (as shown in Fig.6 (b)) LORA exhibits a positive bias across half of the

snow-dominated basins (Fig. 6 (b)) and particularly in Lena, Amur and Yenisei basins  (Fig. 8) .

Pan et al. (2012) and Sheffield et al. (2009) assumed that the errors in the measured streamflow are inversely proportional to the area of the basins and ranges between 5% and 10%. Whereas Di Baldassarre and Montanari (2009) analyzed the overall error affecting streamflow observations and found that these errors range between 6% and 42%. In earlier studies, the errors in streamflow  measurement were estimated to range from 10% to 20%   (Rantz, 1982; Dingman, 1994). In the study of

Zhang et al. (2018), the error ratios of VIC were set to be 5%. In this study, we used the weighting approach to compute gridded uncertainty values based on the discrepancy between the Ragg of the derived runoff and the associated observational dataset in each gauged basin or alternatively, based on the discrepancy between Ragg of the derived runoff and the associated observational dataset from three similar basins in the case of ungauged basins. The derived gridded uncertainty changes in time and space. Our uncertainty estimates show higher values than those set for VIC, and additionally the

estimated values and their reliability change with climate and season (Fig. 7). It follows from Table 2 that in most of the basins the mean annual runoff uncertainty exceeds 30% of the values of the associated runoff itself. In fact, when the values of runoff approach zero (i.e. in arid and semi-arid regions during the hot climate or in the snow dominated basins during winter) it is expected that the uncertainty values become very close to the associated runoff estimates and eventually the error ratio becomes high. It is not surprising that the estimated relative uncertainties exceed the error ratios of the

observations. Also the change of the uncertainty values with time and space is consistent with the fact that the individual datasets that were used to derive LORA exhibit performance differences in different climates and terrains (Beck et al., 2017a).

Figure 9 shows the Mean seasonal runoff (mm/year) calculated for the period 1980 – 2012. There is consistently low runoff in arid regions and high runoff in wet regions across all the seasons. High latitudes in America and Asia exhibit no runoff

during the snow season and high runoff during Mar-Aug when snow melts. Overall, there is a clear agreement between the spatial distribution of runoff and the different climate regimes. This is particularly reflected in Madagascar where the differences in runoff pattern match the different climate regimes across the island. LORA captures the high wetness in the monsoonal seasons and exhibits a shift in magnitude during the wet monsoon in the lower Amazon during Oct-May, the



upper Amazon during Jun-Aug, South Asia during Jun-Nov, Central Sahel in August and Guinea Coasts in June, July, September and October.

As discussed in Hobeichi et al. (2018), the weighting approach has its own advantages and drawbacks. One limitation is that a common imperfection in all the individual products is likely to propagate into the derived product. The early spring runoff peak found in both LORA and the datasets that were used to derive it is an example of this limitation. On the other hand, the seasonal runoff cycle of LORA in both Pechora and Olenek (i.e. two snow-dominated basins) indicate that LORA was able to capture the seasonal signal and the timing of the runoff peak very well as opposed to the constituent products and Best4, which also suggests that the weighting has the ability to overcome the weaknesses of the individual products. Additionally, it was shown in Beck et al. (2017a) that tier1 products consistently overestimate runoff in arid and semi-arid regions due to a bias in the WFDEI precipitation forcing, this appears in the massive overestimation exhibited by Best4 in Niger and Murray-Darling (Fig. 8), however the weighting was able to eliminate a large amount of this overestimation, which also emphasizes the ability of the weighting approach to mitigate limitations in individual models. Another limitation arises from the scarcity of observed streamflow particularly in the arid regions and from the quality of the observational data itself.  As noted earlier, the errors in GRDB dataset were reported to range between 10% and 20% and were found by Di Baldassarre and Montanari (2009) to have an average value that exceed 25% across all the studied river basins.

The weighting technique allows the addition of new runoff estimates when they become available. This will be particularly beneficial if the future estimates represent reasonably the runoff peak in the snow-dominated regions.

## 5 Conclusion

In this study, we presented LORA, a new global monthly runoff product with associated uncertainty. LORA was derived for 1980–2012 with monthly temporal resolution at 0.5° spatial resolution by applying a weighting approach that accounts for both performance differences and error covariance between the constituent products.

To ensure full global coverage, we used a similarity index to transfer weights and bias ratios constructed from gauged basins with similar climatic and physiographic characteristics to ungauged basins. This allows the derivation of runoff in areas where we do not have observed streamflow.

We showed that this approach is succeeding, that LORA performs better than any of its constituent modelled products in a range of metrics, across basins globally and especially in the higher latitudes. However, LORA tends to overestimate runoff and shows an early snow-melt peak in some snow-dominated basins. LORA was not found to significantly overestimate runoff in arid and semi-arid regions as opposed to the constituent products.

The approach and product detailed here offers the opportunity for improvement as new streamflow and modelled runoff datasets become available. It presents a new, relatively independent estimate of a key component of the terrestrial water budget, with a justifiable and well constrained uncertainty estimate.



## 6 Competing interests

The authors declare that they have no conflict of interest.

## 7 Acknowledgment

Sanaa Hobeichi acknowledges the support of the Australian Research Council Centre of Excellence for Climate System Science (CE110001028). Gab Abramowitz and Jason Evans acknowledge the support of the Australian Research Council Centre of Excellence for Climate Extremes (CE170100023). Hylke Beck was supported by the U.S. Army Corps of Engineers' International Center for Integrated Water Resources Management (ICIWaRM), under the auspices of UNESCO. This research was undertaken with the assistance of resources and services from the National Computational Infrastructure (NCI), which is supported by the Australian Government. We are grateful to the Global Runoff Data Centre (GRDC) for providing observed streamflow data. We thank the participants of the eartH2Observe project for producing and making available the model simulations. We also acknowledge that the HydroBASINS product has been developed on behalf of World Wildlife Fund US (WWF), with support from, and in collaboration with: the EU BioFresh project, Berlin, Germany; the International Union for Conservation of Nature (IUCN), Cambridge, UK; and McGill University, Montreal, Canada. Major funding for this project was provided to WWF by Sealed Air Corporation; additional funding was provided by BioFresh and McGill University.

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

**Tables**

**Table 1: Model outputs from Tiers 1 and 2 of eartH2Observe project used to derive the synthesis runoff product.**

| Model | Tier | Our abbreviation | Variables | Spatial Resolution | Reference |
|---|---|---|---|---|---|
| | | | | | |



| HTESSEL | 1 | HTESS1 | Streamflow & Total runoff | 0.5° | (Balsamo et al., 2009, 2011) |
|---|---|---|---|---|---|
| | 2 | HTESS2 | streamflow & Total runoff | 0.25° | (Balsamo et al., 2009, 2011) |
| JULES | 1 | JULES1 | Total runoff | 0.5° | (Best et al., 2011) |
| | 2 | JULES2 | Total runoff | 0.25° | (Best et al., 2011) |
| LISFLOOD | 1 | LISF | Streamflow & Total runoff | 0.5° | (Burek, P., van der Knijff, J., de Roo, 2013; Van Der Knijff et al., 2010) |
| PCR-GLOBWB | 1 | PCRG | Streamflow & Total runoff | 0.5° | (Van Beek and Bierkens, 2009) |
| SURFEX | 1 | SURF1 | Streamflow & Total runoff | 0.5° | (Decharme et al., 2011, 2013) |
| | 2 | SURF2 | Total runoff | 0.25° | (Decharme et al., 2011, 2013) |
| W3RA | 1 | W3RA | Streamflow & Total runoff | 0.5° | (Van Dijk et al., 2014; Van Dijk and Warren, 2010) |
| WaterGAP3 | 1 | WGAP3 | Streamflow & Total runoff | 0.5° | (Flörke et al., 2013) |
| HBV-SIMREG | 1 | HBVS | Total runoff | 0.5° | (Beck et al., 2016) |

**Table 2: A comparison of mean annual runoff (mm/year) of 16 major basins covering different climate zones around the world for LORA and VIC (Zhang et al., 2018), the mean annual uncertainty values associated with LORA runoff are shown and the adjusted VIC annual runoff values within 5% error bounds for water budget closure are displayed.**

| Basin | VIC mm/year | VIC adjusted for water budget closure mm/year | LORA (Runoff) mm/year | LORA (uncertainty) mm/year | Dominant climate |
|---|---|---|---|---|---|
| Amazon | 1048 | 1029 | 1151 | 357 | Tropical wet |
| Amur | 135 | 129 | 219 | 115 | Humid continental and semi arid |
| Columbia | 318 | 293 | 333 | 101 | Semi-arid and highlands |
| Congo | 407 | 404 | 358 | 147 | Tropical wet and tropical dry |
| Danube | 272 | 265 | 260 | 125 | Marine Humid, continental and humid subtropical |



| | | | | |
|---|---|---|---|---|
| Indigirka | 132 | 120 | 228 | 171 | Subarctic |
| Lena | 142 | 134 | 301 | 137 | Subarctic |
| Mackenzie | 189 | 173 | 191 | 110 | Subarctic |
| Mississippi | 220 | 215 | 212 | 123 | Humid continental and humid subtropical |
| Murray-Darling | 42 | 41 | 15 | 6 | Arid and semi-arid |
| Niger | 198 | 194 | 106 | 41 | Arid, semi-arid and tropical dry |
| Olenek | 114 | 106 | 230 | 208 | Subarctic |
| Parana | 278 | 279 | 189 | 97 | Marine and humid subtropical |
| Pechora | 342 | 308 | 420 | 420 | Tundra and subarctic |
| Yenisei | 217 | 195 | 324 | 203 | Subarctic |
| Yukon | 149 | 139 | 229 | 102 | Subarctic |

**Figures**

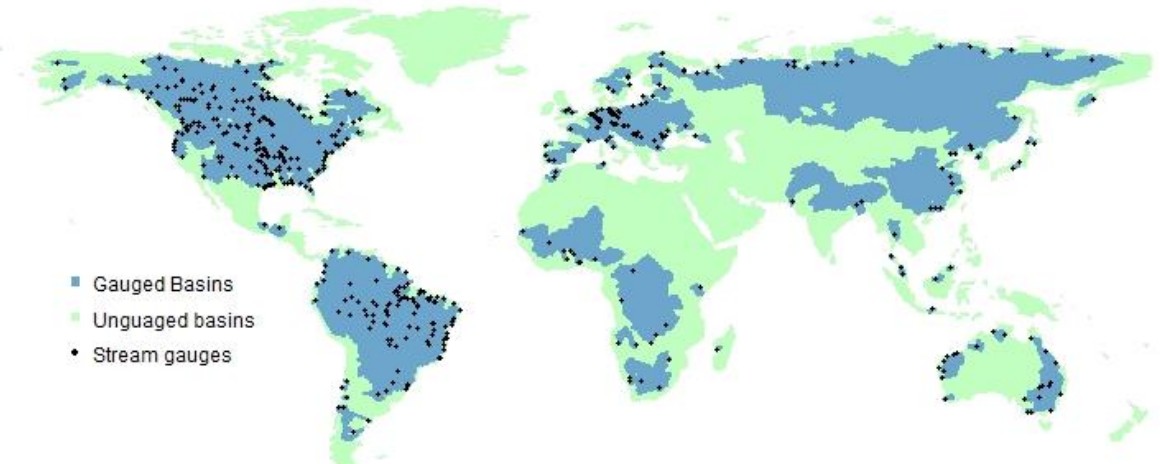

5    **Figure 1: Spatial coverage of gauged and ungauged river basins and location of stream gauges.**



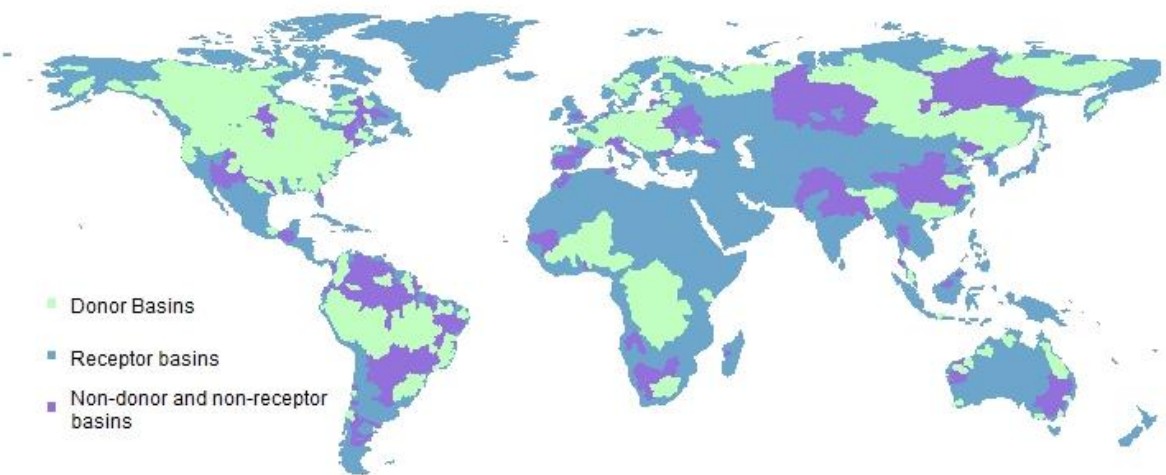

**Figure 2: Spatial coverage of donor basins, receptor basins and non-donor and non-receptor basins.**

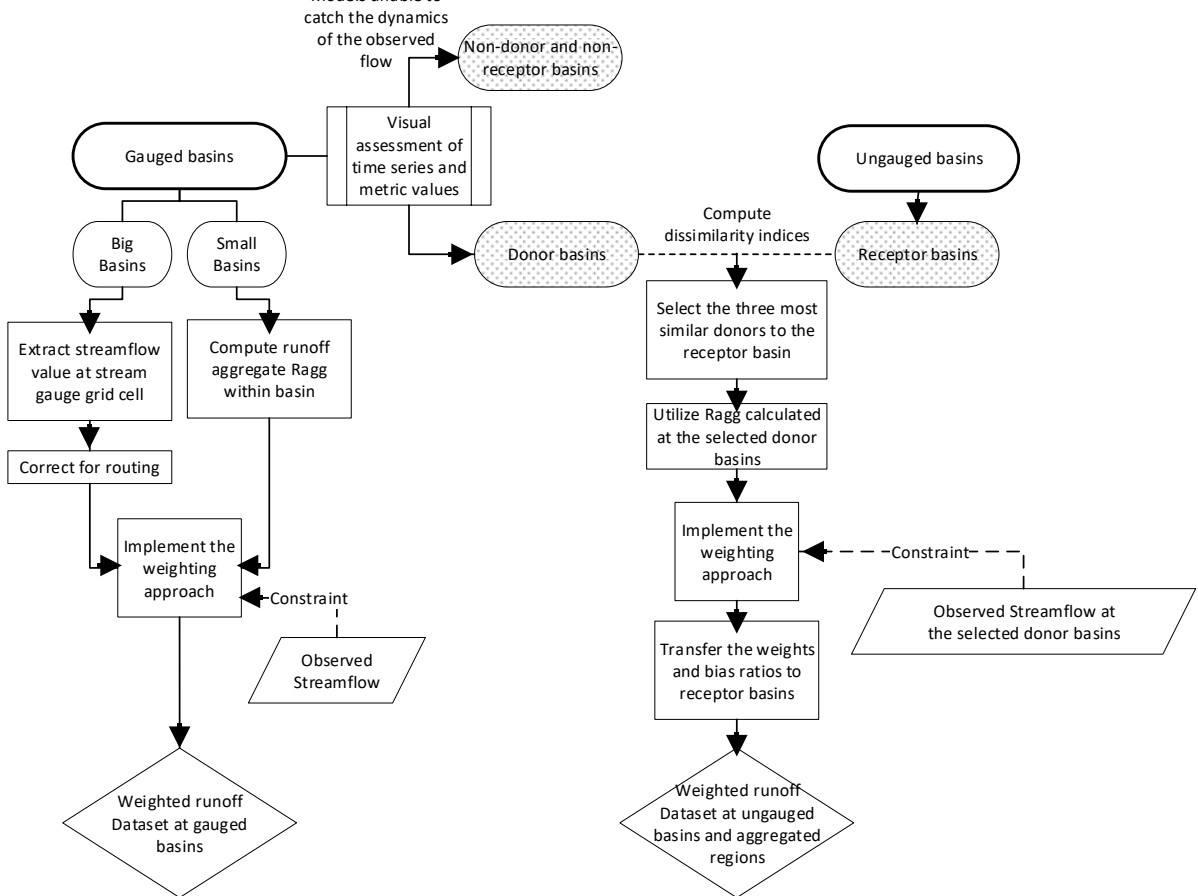

**Figure 3. Flow chart summarizing the steps carried out to derive the weighted runoff product for the global land surface.**



(a)

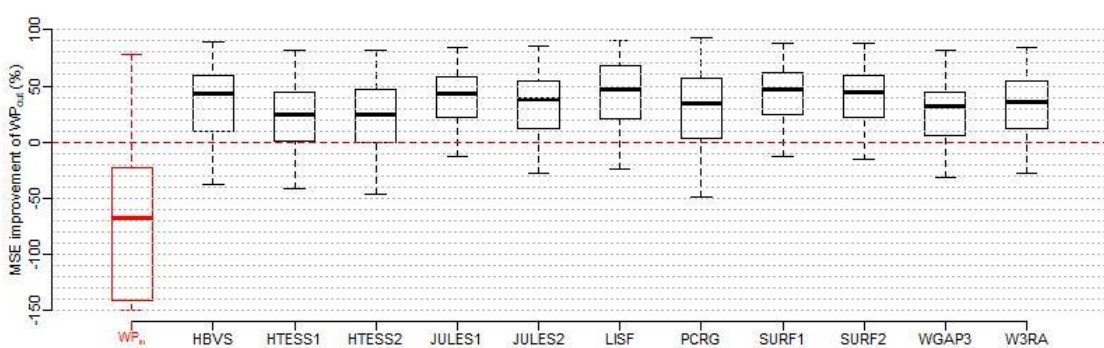

(b)

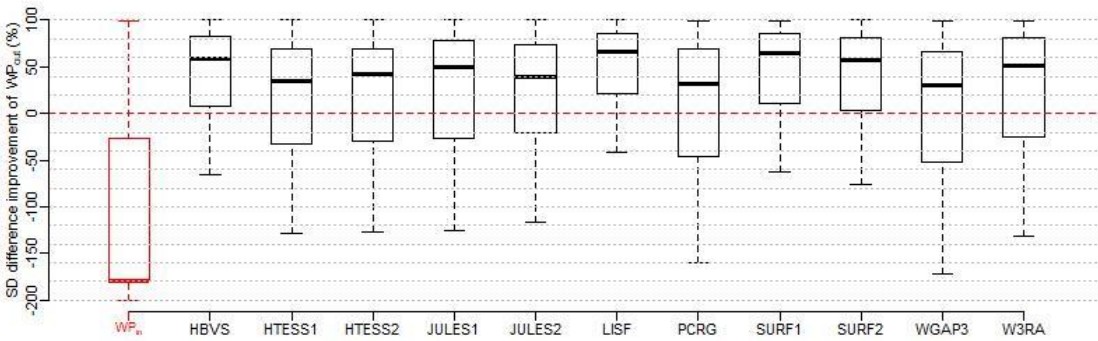

(c)

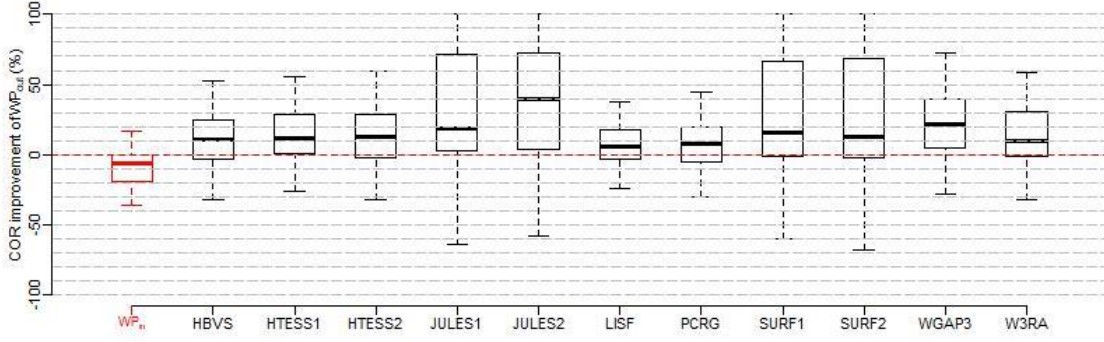

(d)

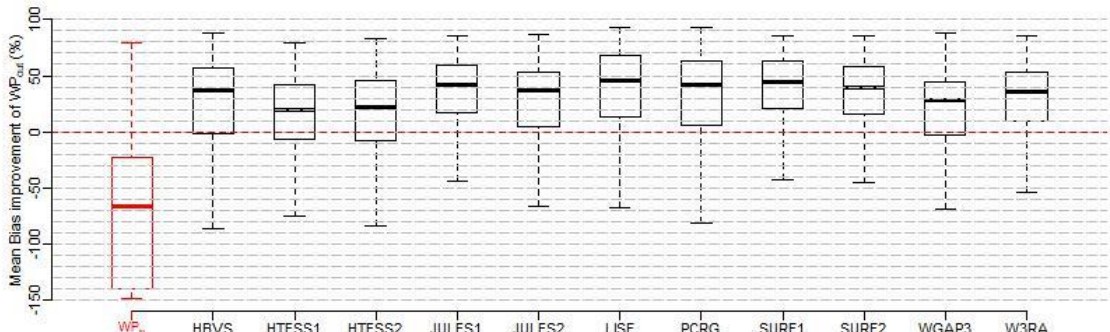





**Figure 4: Box and whisker plots displaying the percentage improvement that the weighted product ($WP_{out}$) offers when tested out-of-sample, using four metrics: MSE (a), SD difference (b), COR (c) and Mean bias (d), when compared to the weighted product derived from in-sample data ($WP_{in}$), and each runoff product involved in this study. Box and whisker plots represent values calculated at 482 gauged basins. See Table 1 for dataset abbreviations. The lower and upper hinges of a boxplot represent the first ($Q_1$) and third ($Q_3$) quartiles respectively of the performance improvement results and the line inside the boxplot shows the median value. The extreme of the lower whisker represents the maximum of 1) min(dataset) and 2) ($Q_1$ - IQR), while the extreme of the upper whisker is the minimum of 1) max(dataset) and 2) ($Q_3$ + IQR)), where IQR represents the interquartile range (i.e. $Q_3$ - $Q_1$ ) of the performance improvement results. A median line located above the 0 axis is an indication that the out of sample weighting offers an improvement in more than half of the basins.**



(a)

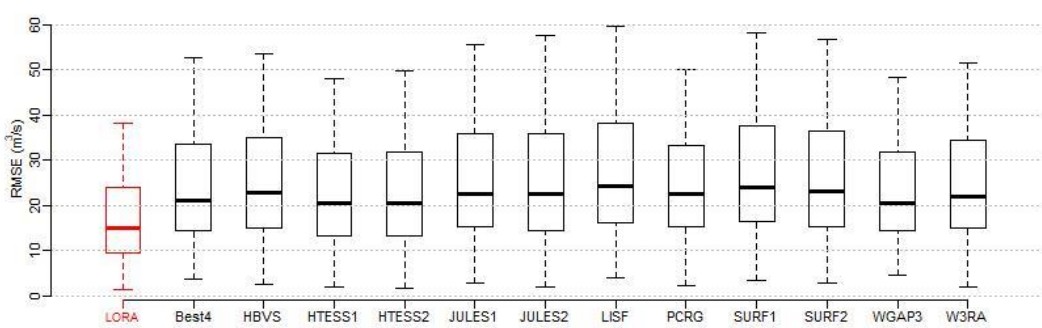

(b)

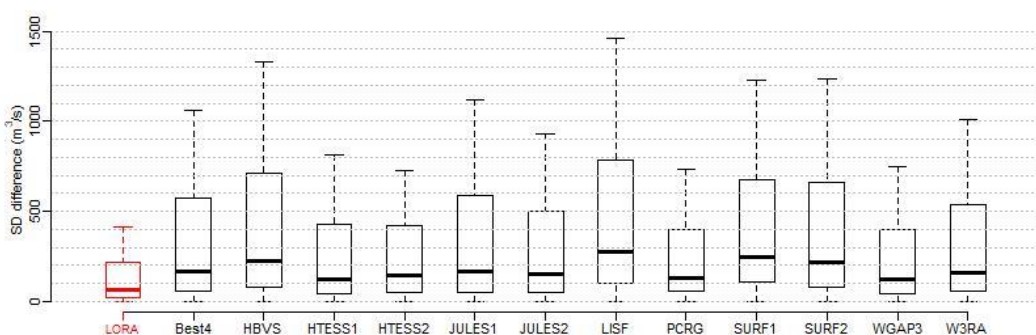

(c)

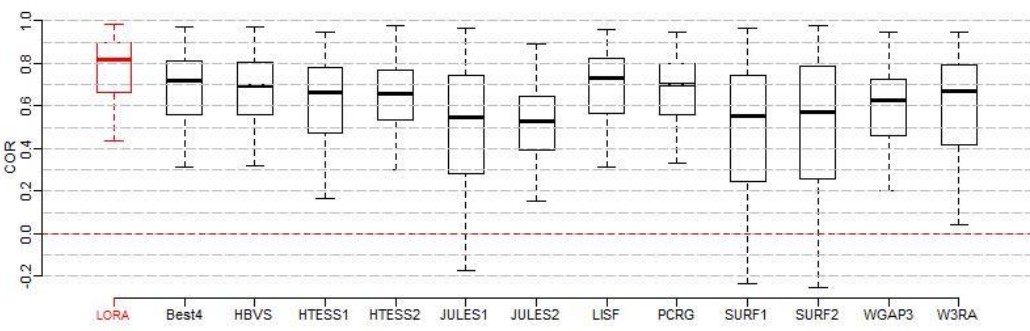

(d)

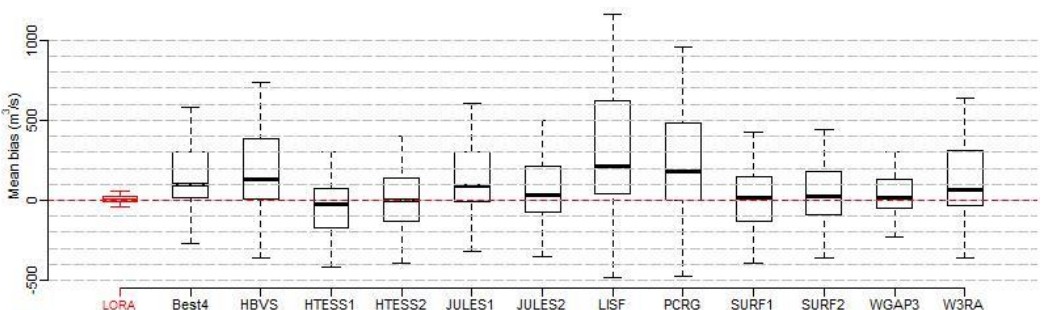



**Figure 5: Four statistics, (a) RMSE, (b) SD difference, (c) COR and (d) Mean bias, calculated for LORA, Best4 (i.e. the simple average of runoff estimates from LISFLOOD, WaterGAP3, W3RA and HBV-SIMREG) and each runoff product involved in this study at the gauged basins. See Table 1 for dataset abbreviations.**

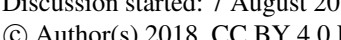

**Figure 6: Two statistics, (a) COR and (b) Mean bias, calculated for LORA, Best4 (i.e. the simple average of runoff estimates from LISFLOOD, WaterGAP3, W3RA and HBV-SIMREG) and each runoff product involved in this study at the gauged basins located at the high latitudes (>60°). See Table 1 for dataset abbreviations.**



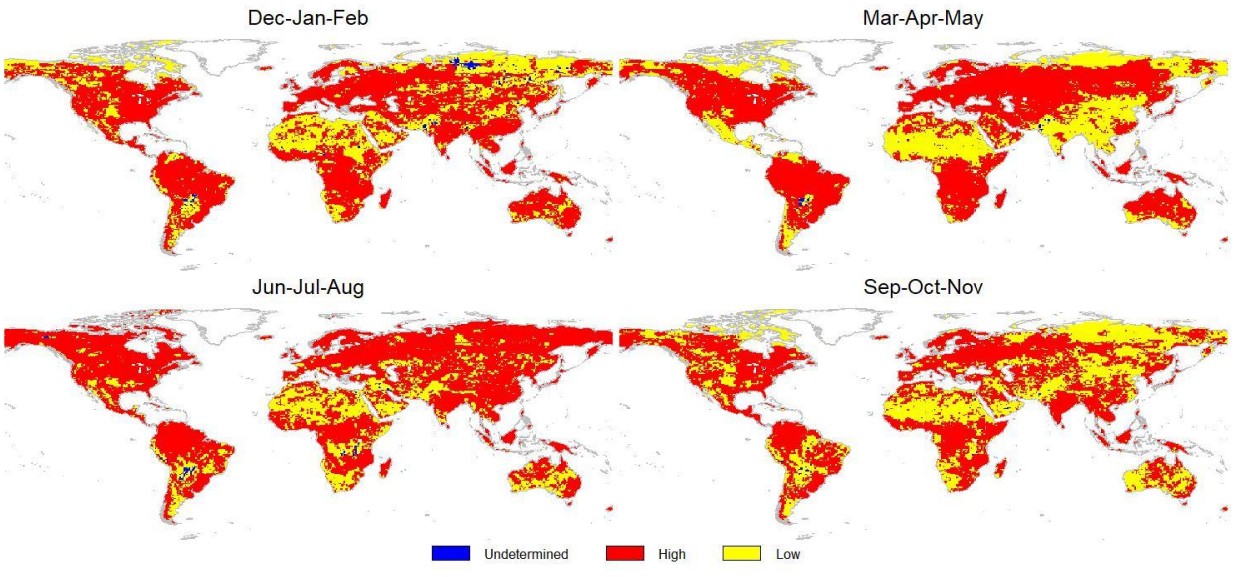

**Figure 7: Seasonal reliability, defined as** high ($\frac{men\ runoff\ uncertainty}{mean\ runoff} < 1$, **in red), low** ($\frac{men\ runoff\ uncertainty}{mean\ runoff} \geq 1$, **in yellow) and undetermined (mean runoff = 0, in blue).**



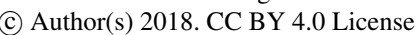

**Figure 8: Seasonal cycle of Runoff aggregates from LORA and Best4 compared with the observed streamflow over 11 major basins. Runoff aggregates and the observed streamflow were averaged for each month across the period of availability of observation. The shaded regions shows the aggregated uncertainty derived for LORA.**



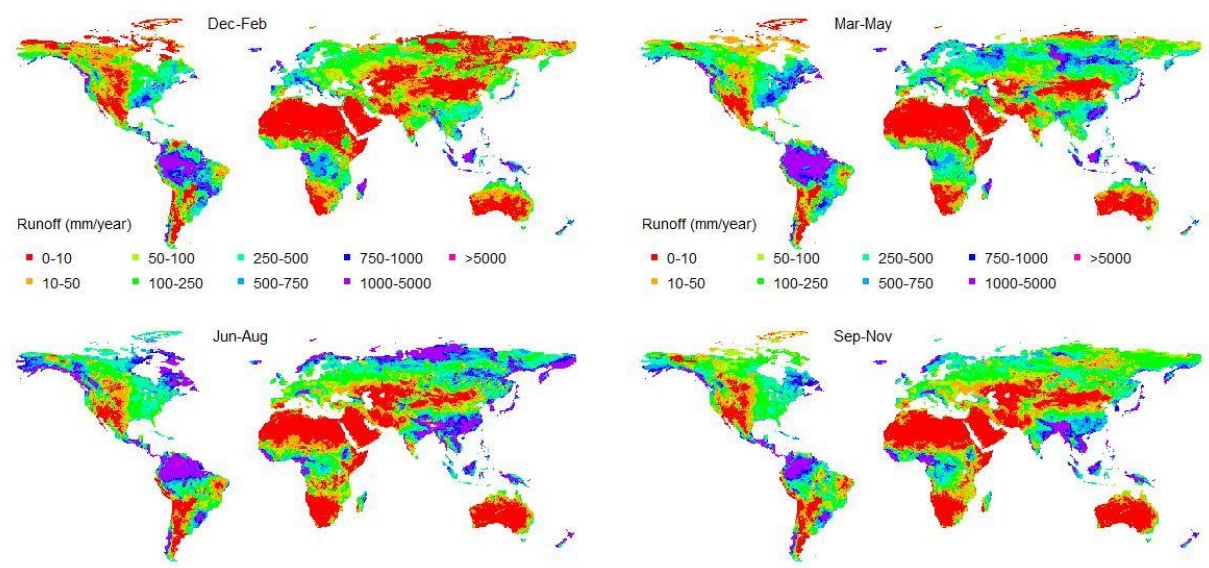

**Figure 9: Mean seasonal runoff calculated for the period 1980 – 2012**