# Peer review of "Linear Optimal Runoff Aggregate (LORA): A global gridded synthesis runoff product"

_Hydrology and Earth System Sciences, 2018_

## Referee Comment (RC1) · L. Gudmundsson (Referee) · 7 Sep 2018

I did enjoy reading the paper of Hobeichi and colleagues, which describes an interesting approach to compute a weighted ensemble mean of runoff fields, conditional on observed river flow. Overall the paper is well written. Nevertheless, I do have some comments and requests for clarification as summarized below:

** Overall comments and questions **

(1) Please refer early to Fig. 3 in the methods description.

(2) Maps of the performance of LORA would be very helpful.

(3) Some example time series, including the time-varying uncertainty would be interesting.

(4) Does the fact that the weights are computed based on a discrete set of donor-catchments lead to discontinuities in the runoff fields? How would e.g. the field look like for one individual month?

(5) I really appreciate the authors effort to also include uncertainty estimates in their product. Unfortunately, I did not find any validation of this uncertainty estimate or a full interpretation of what it means. I know the uncertainty estimates are introduced elsewhere, but as this is a relatively new approach it requires extra care. For example:

(5.1) It is not clear to me whether this uncertainty estimate is related to the "confidence interval" (i.e. an estimate for the range of the "true" ensemble mean") or whether it is related to the "prediction interval" (i.e. an estimate of the range in which new observations would fall).

(5.2) if the uncertainty is an "prediction interval", an evaluation of the widths of the "uncertainty bounds" with respect to the distribution of the residuals would be helpful. Especially, compared to the spread of the input ensemble. For this, metrics from ensemble forecasting (e.g. the continuous ranked probability score (CRPS) or "reliability plots/histograms" might be helpful).

(5.3) the uncertainty bounds can produce negative runoff. Should this be the case or is this an artefact?

(6) I would appreciate some more information (figures, tables) on the actual "weights" and "biases"

(7) some of the references have artefacts, e.g. "nan" values instead of page numbers. . .

** Specific comments and questions **

Section 2.1: The authors might also be interested in the following global-scale data source: Do et al (2018, doi:10.5194/essd-10-765-2018) & Gudmundsson et al (2018,

doi: 10.5194/essd-10-787-2018). Sorry for this self-citation.

Section 2.2: What about other comprehensive model ensembles, such as ISIMIP2a (http://dx.doi.org/10.5880/PIK.2017.010)

page 4, line 25: Is there some empirical evidence supporting the assumption of constant bias ratio?

Generall for methods: I found the mixed use of Ragg_kˆj and q_kˆj confusing. Would it be possible to clarify the difference/commonality once and then only use one of them?

page 5, line 2: not really clear what you mean with this sentence. what is the number of records?

page 5, line 4: An alternative approach for dealing with negative values might be to log-transform runoff/stream flow before doing the computations and back transform it in the end. See e.g. Gudmundsson & Seneviratne (2015, doi: 10.5194/hess-19-2859-2015; 2016, doi: 10.5194/essd-8-279-2016). Again, my apologies for self-citation.

page 5, line 10: Some more details on the transformation process would be appreciated. I know it is published in Hobeichi et al. (2018) but it would make the paper easier to understand if it would be outlined in more detail here.

page 5, line 20: transfer of "weights" from the 3 most similar basins; why 3 and not a larger/smaller number? Any empirical motivation for this? Would optimizing this hyper-parameter help to get even better results?

page 6. line 8: Note that "Tundra" and "Subarctic" will not always have permanent ice! Also: The chosen climate zone map is very uncommon, and most readers will not be familiar with it. Therefore, it needs to be presented in a figure. Alternative: why not use a common climate-zone definition (e.g. Köppen-Geiger?)

page 7. l 15: should be fig 4 not fig 3

Fig 4: I found the "relative" improvement difficult to grasp upon first reading. Suggestion: just show the performance for WPin WPout (and omit individual models, as they are shown later)

page 9, line 18:

(1) "mean runoff uncertainty". What was exactly calculated? Note that simply averaging sqrt(sigma) will yield wrong results. Instead the rules for propagation of uncertainty should be used (https://en.wikipedia.org/wiki/Propagation_of_uncertainty). If the authors did already consider this, I apologize for this comment.

(2) Note that there are strong perceptions on what "reliability" means in ensemble forecasting. See e.g. "reliability diagrams" and "reliability histograms"

page 9, line 31 & Fig 8: how did you compute the mean uncertainty? (refer to my comment above)

page 10, line 31: not Fig 3...

―――――――――――――――

---

## Referee Comment (RC2) · Anonymous Referee #2 · 2 Nov 2018

**Review of "Linear Optimal Runoff Aggregate (LORA): A global gridded synthesis runoff product"**
**by Hobeichi et al.**

The paper presents a new global gridded dataset at half degree resolution based on a weighted average of global runoff and discharge estimates from global hydrological models constrained by discharge observations. It represents a valuable new dataset that may be of interest to hydroclimatologists or environmental scientists. The paper is concise and well written and deserves to be published in HESSD. There are a few issues that need to be resolved however.

1. I miss a few explicit examples explaining why runoff is at all useful, especially at 0.5 degrees. The argument is that accurate estimates of runoff are critical to inform climate change adaptation strategies, to guide appropriate water management in agriculture and to enable the assessment of the impact of anthropogenic activities on ecosystems. However, what does runoff at 50x50 km resolution even mean? It is in terms of scale too far off from being operationally relevant. Thus, a stronger justification using examples is called for.

2. I also miss some references to the earliest work on runoff fields, e.g. Fekete et al. 2002: https://agupubs.onlinelibrary.wiley.com/doi/full/10.1029/1999GB001254

3. Regarding to previous work and scales, I would like to call the authors' attention to a recent publication by Barbarossa et al.( https://www.nature.com/articles/sdata201852). They provide discharge estimates at 1 km resolution. I move that these are probably more representative for local runoff than those obtained from GHMs at half degree resolution?

4. Section 2.3: I find that there is too little info on the method used to compute the weights. I don't think that it should be necessary read another paper to comprehend the essentials of the methods used. So, I would want some more explanation on how the weights are calculated.
   For instance:
   - How are correlations between models accounted for?
   - Are the weights allowed to be negative?
   - Is the sum of the weights adding up to one? If this is the case, one has to add another equation and transform a constrained optimization to an unconstrained one using Lagrange multipliers.

5. Line 8-10 page 5: This seems to assume that travel times are less than a month because it neglects routing?

6. Lines 11-13 page 5: "It provides better estimates than simply calculating the standard deviation of the involved products". Is that really the case? If your weighting method is optimal and you have bias-corrected correctly would the following estimator for each pixel not be unbiased (i and j are different products bias corrected):

$$\hat{\sigma}_r^2 = \frac{1}{N \times 11 \times 11} \sum_{i=1}^{11} \sum_{j=1}^{11} \sum_{t=1}^{T} w_i w_j (r_{i,t} - \mu_r)(r_{j,t} - \mu_r)$$

   By moving a window T over time you get your time varying variance

7. Line 20-25 page 5: transferring the weights from donors to receptors. Are one set of weights obtained jointly for the three donor catchments or are three sets of weights averaged and transferred?

8. Line 11 on page 10. Why compare only with VIC? Why not GLDAS (4 models).

9. Table 2: why not add the estimated total runoff volumes from GRDC (also globally in km$^3$). It would be good to see what the global runoff volume is in this product compared to other estimates.

10. Line 27 on page 10: reduced performance in dry climates. Apart from the reasons mentioned, another possible cause could be the fact that GHMs are probably less proficient in representing runoff processes in arid basins where often runoff is local and will not always be turned into streamflow

---

## Author Comment (AC1) · 28 Nov 2018

**Manuscript hess-2018-386 entitled "Linear Optimal Runoff Aggregate (LORA): A global gridded synthesis runoff product"**

We would like to thank Lukas Gudmundsson for his constructive comments on our manuscript. This document outlines our responses to his comments and the improvements made to the manuscript.

**Response to overall comments and questions**

**(1) Please refer early to Fig. 3 in the methods description.**
Thanks for this suggestion, we have now referred to the flowchart in Figure 3 before detailing our methods. Its caption has now changed to Figure 2.

**(2) Maps of the performance of LORA would be very helpful.**

**(3) Some example time series, including the time-varying uncertainty would be interesting.**

Both (2) and (3) are good ideas. We have now provided a map of the temporal correlation of aggregated LORA runoff and observed streamflow, as well as time series computed over selected basins with different correlation levels, showing LORA uncertainty estimates. We have also added the text:

> *Further, we provide in Fig. S2 the spatial distribution of correlation results from Fig. 6 (c). The basins are colour-coded by their temporal correlation with the observed streamflow and the number of basins in each category is given. Basins in yellow are those where LORA is highly correlated with the observation while dark blue basins are those where LORA exhibits a negative correlation with the observation. It can be noted from Fig. 6(c) that occurrence of negative correlation is extremely unusual which explains why these were considered outliers and were not shown in the box and whisker plot. Likely, low correlation basins are unusual and constitute less than 12% of the number of basins (excluding basins with negative correlation). Also, the median value is above 0.8, which is higher than any constituent estimates. We selected a basin from each correlation range and examined the timeseries of LORA and the observed streamflow more closely (Fig. S3-S7), in particular illustrating the uncertainty estimate of LORA. In Ganges, LORA captures well the observed time-series dynamic with a tendency to over-estimate streamflow peak in August (Fig. S3). Over Madeira basin, LORA is able to represent reasonably well most of the climatic variability found in the observation (Fig. S4). In Congo, the catchment has an irregular time-series dynamic, LORA is in principle able to capture a large part of the climatic variability in the observation (Fig. S5). In Lena, the observation shows a peak in June and a second less significant peak in September (Fig. S6). Both peaks are captured by LORA during most of the time series with a tendency to underestimate the late summer peak and overestimate the early summer peak. In the upper Indus, LORA does not capture the magnitudes of observed streamflow and shows a reversed seasonal cycle which explains why it exhibits negative*

*correlation with the observation (Fig.S7). Zhang et al. (2018) found disagreement between simulated runoff from three LSMs and observed streamflow over Indus basin which they expected to be due to errors in the observational data from GRDB dataset.*

[Figure]

*Figure S2: Temporal correlation of LORA with the observed streamflow over the gauged basins. Basins are colour coded by correlation range and their numbers are given in brackets.*

[Figure]

*Figure S3: observed streamflow (in black), LORA Runoff aggregate (in red), and its uncertainty range (grey) over the Ganges basin (in mm month$^{-1}$). This basin was shown in yellow in Fig. S2, indicating that LORA exhibits a high temporal correlation ($\geq$ 0.9) with the observation.*

[Figure]

*Figure S4: observed streamflow (in black), LORA Runoff aggregate (in red), and its uncertainty range (grey) over Madeira basin, i.e. a sub-basin of the Amazonas (in mm month$^{-1}$). This basin was shown in orange in Fig. S2, indicating that LORA exhibits a temporal correlation in the range [0.75 – 0.9[ with the observation.*

[Figure]

*Figure S5: observed streamflow (in black), LORA Runoff aggregate (in red), and its uncertainty range (grey) over the Congo basin (in mm month$^{-1}$). This basin was shown in violet in Fig. S2, indicating that LORA exhibits a temporal correlation in the range [0.5 – 0.75[ with the observation.*

[Figure]

*Figure S6: observed streamflow (in black), LORA Runoff aggregate (in red), and its uncertainty range (grey) over Lena basin (in mm month$^{-1}$). This basin was shown in purple in Fig. S2, indicating that LORA exhibits low temporal correlation (<0.5) with the observation.*

[Figure]

*Figure S7: observed streamflow (in black), LORA Runoff aggregate (in red), and its uncertainty range (grey) over the Indus basin (in mm month$^{-1}$). This basin was shown in dark blue in Fig. S2, indicating that LORA exhibits a negative temporal correlation with the observation.*

**(4) Does the fact that the weights are computed based on a discrete set of donor catchments lead to discontinuities in the runoff fields? How would e.g. the field look like for one individual month?**

Good point. We obviously don't expect to see discontinuity in the runoff fields within individual gauged basins since the weighting is uniform across the basin. On the other hand, we expect to see discontinuity in runoff fields at ungauged basins, particularly over large heterogeneous ungauged basins, given that different sets of weights were used to derive runoff over different parts of the basins. However, since dividing the heterogeneous basins into different regions was based on climatic differences between these regions, we think that such discontinuity naturally arises from the differences in climatic characteristics anyway, so that a different set of weights for each region is not necessarily problematic. It certainly seems a better use of available information that weighting globally. A visual assessments of LORA monthly maps did not reveal unexpected discontinuities in the runoff fields. We provide in Fig. S8 an example of runoff fields in an individual month (e.g. May, 2003). Since LORA is publicly available it should be easy to verify that this example is not a misrepresentation of the results.

[Figure]

**Fig. S8: Global map of LORA runoff fields (mm) in May 2003.**

**(5) I really appreciate the authors effort to also include uncertainty estimates in their product. Unfortunately, I did not find any validation of this uncertainty estimate or a full interpretation of what it means. I know the uncertainty estimates are introduced elsewhere, but as this is a relatively new approach it requires extra care.**

We have now explained in more detail how we calculated the uncertainty estimates:

*We implemented the ensemble dependence transformation process detailed in Bishop and Abramowitz (2013) to compute the gridded time-variant uncertainty associated with the derived runoff estimates. For any given gauged basin, we first calculated the spatial aggregate of our weighted runoff estimate $Ragg_\mu$, then quantified $s_q{}^2$, the error variance of $Ragg_\mu$ with respect to the observed streamflow Q over time as:*

$$s_q{}^2 = \frac{\sum_{j=1}^{J}(Ragg_\mu{}^j - Q^j)^2}{J-1}$$

*Then, we wished to guarantee that the variance of the constituent modelled estimate $\sigma_q^{2\,j}$ about $Ragg_\mu{}^j$ at a given time step, averaged over all time steps where we have available streamflow data, is equal to $s_q{}^2$, such as $s_q{}^2 = \frac{1}{J}\sum_{j=1}^{J}\sigma_q^{2\,j}$.*

*Since the variance of the existing constituent products do not, in general, satisfy this equation. We transformed them so that it does. This involved first modifying the set of weights w to a new set $\widetilde{w}$ such that*

$\widetilde{w} = \frac{w^T + (\alpha - 1)\frac{1^T}{K}}{\alpha}$, *where $\alpha = 1 - K min(w_k)$ and $min(w_k)$ is the smallest negative weight (and $\alpha$ is set 1 if all $w_k$ are non-negative). This ensures that all the modified weights $\widetilde{w}_k$ are positive. We then transform the individual estimates $x_k^j$ to $\tilde{x}_k^j$ where $\tilde{x}_k^j = Ragg_\mu{}^j + \beta(\bar{x}^j + \alpha(x_k^j - \bar{x}^j) - Ragg_\mu{}^j)$ and $\beta =$*

$\sqrt{\dfrac{s_q^2}{\frac{1}{J}\sum_{j=1}^{J}\sum_{k=1}^{K}\widetilde{w}_k(\bar{x}^j + \alpha(x_k^j - \bar{x}^j) - Ragg_\mu{}^j)^2}}.$

*The weighted variance estimate of the transformed ensemble can be defined as*

$\sigma_q^{2\,j} = \sum_{k=1}^{K}\widetilde{w}_k(\tilde{x}_k^j - Ragg_\mu{}^j)^2$ *and ensures that the equation $\frac{1}{J}\sum_{j=1}^{J}\sigma_q^{2\,j} = s_q^2$ holds true. Furthermore, $\sqrt{\sigma_q^{2\,j}}$ is the temporally varying estimate of uncertainty standard deviation of the transformed ensemble that (a) is varying in time, and (b) accurately reflects our ability to reproduce the observed streamflow.*

*We refer the reader to Bishop and Abramowitz (2013) for proofs.*

*In order to estimate $\sqrt{\sigma_r^{2j}}$ , the uncertainty of the runoff attributes $\mu_r^j$ at each point in time and space, we first transformed the runoff fields $r_k^j$ to $\tilde{r}_k^j$ by applying the same transformation parameters $\alpha$ and $\beta$ such that $\tilde{r}_k^j = \mu_r^j + \beta(\bar{r}^j + \alpha(r_k^j - \bar{r}^j) - \mu_r^j)$. We then calculated the error variance $\sigma_r^{2j} = \sum_{k=1}^{K} \tilde{w}_k(\tilde{r}_k^j - \mu_r^j)^2$.*

*Finally, we used $\sqrt{\sigma_r^{2j}}$ as the spatially and temporally varying estimate of runoff uncertainty standard deviation, which we will refer to below simply as 'uncertainty'. It provides a much more defensible uncertainty estimate than simply calculating the standard deviation of the involved products.*

*We note that for a given basin, $\sqrt{\sigma_q^{2j}}$ represents the uncertainty of the modelled streamflow i.e. $Ragg_\mu{}^j$, while $\sqrt{\sigma_r^{2j}}$ represents the uncertainty of modelled runoff at each grid cell across the basin. This means that at every time step, there is one value for $\sqrt{\sigma_q^{2j}}$ per basin, and one value for $\sqrt{\sigma_r^2}$ per grid across the basin.*

**(5.1) For example: It is not clear to me whether this uncertainty estimate is related to the "confidence interval" (i.e. an estimate for the range of the "true" ensemble mean") or whether it is related to the "prediction interval" (i.e. an estimate of the range in which new observations would fall).**

(5.1) It follows from the above that $\sqrt{\sigma_q^{2j}}$ can be considered a 'prediction interval' if it was calculated over an ungauged basin. So $\sqrt{\sigma_r^{2j}}$ , the uncertainty estimate associated to the runoff fields, is also considered a prediction interval. The applied transformation can be viewed as a tool to extrapolate the uncertainty estimates in two directions: 1) basin-to-basin: from gauged to ungauged basins, and 2) basin-to-grid: from runoff aggregate in a basin to grid cells across the same basin.

**(5.2) if the uncertainty is an "prediction interval", an evaluation of the widths of the "uncertainty bounds" with respect to the distribution of the residuals would be helpful. Especially, compared to the spread of the input ensemble. For this, metrics from ensemble forecasting (e.g. the continuous ranked probability score (CRPS) or "reliability plots/histograms" might be helpful).**

We don't really have observations for runoff, so we can't test our method for deriving uncertainty on runoff, however we can test it on streamflow (i.e. runoff aggregates). At the gauged basins, our method for deriving the uncertainty estimates guarantees -by design- that the

uncertainty estimates are equal to the RMSE of runoff aggregates against observed streamflow. Therefore, we see that the good performance of uncertainty estimates at the gauged basins is obvious. However, to test that our approach is also succeeding over the ungauged basins, we have now performed out-of-sample tests to show that the distribution of the errors over the gauged basin is similar to the distribution of their errors when they are considered ungauged. We have now explained how we have performed this test and showed the results in the manuscript:

> *The uncertainty estimates computed at the gauged basins represent the deviation of (the spatial aggregate of) our weighted product ($Ragg_\mu$) from the observed streamflow, since the in-sample uncertainty estimates are calculated from the variance of the transformed ensemble, which by design equals MSE of $Ragg_\mu$ against observations (i.e. error variance of $Ragg_\mu$). To test if the uncertainty estimates perform well out-of-sample (i.e. at the ungauged basins), we took a gauged basin, but instead of constraining the weighting using observed streamflow from this basin, we constructed model weights by using the three most similar donor basins. We could then calculate MSE of $Ragg_\mu$ against observations from the three donor basins, denoted by MSE$_{in}$, which provides us with the uncertainty estimates calculated in-sample ($\sqrt{MSE_{in}}$), since the observational data used in this case is the same dataset that was used to train the weighting. We also calculated the MSE of the aggregated weighted product against the actual observation of the gauged basin and denoted this MSE$_{out}$. $\sqrt{MSE_{out}}$ represents the uncertainty estimates computed out-of-sample, since the comparison was performed against observational data that has not been used to train the weighting. We repeated the out-of-sample test for all the gauged basins.*

> *We displayed the results of the out-sample-test by showing the ratio $\sqrt{MSE_{out}} / \sqrt{MSE_{in}}$. If the approach is succeeding, we expect that this ratio is around one, indicating that the values of MSE$_{in}$ and MSE$_{out}$ are close to each other. We used a box and whisker plot, where each sample is a different basin, to show the results.*

We have also commented on the results:

> Critically though, the fact that the weighting delivers improvement over all models when the weights are transferred from similar basins indicate that the dissimilarity technique is succeeding and can be effectively used at the ungauged basins by feeding the weighting with data from the most similar basins with streamflow observations. *Furthermore, the boxplot in Fig 5 shows that, overall, when the uncertainty estimates are computed out-of-sample they are very similar to what they would have been if they were computed in-sample. This demonstrates that the dissimilarity technique can be effectively used to derive not only the weighting product but also its associated uncertainties at the ungauged basin.*

[Figure]

*Figure 5: Box and whisker plots displaying the ratio of (1) the uncertainties of the spatial aggregate of the weighted product computed out-of-sample to (2) the uncertainties of the spatial aggregate of the weighted computed in-sample.*

**(5.3) the uncertainty bounds can produce negative runoff. Should this be the case or is this an artefact?**

Good point. When runoff is smaller than its associated uncertainty, the uncertainty bounds will certainly produce negative runoff. This is an artefact that arises from using a generic process to determine an uncertainty range. A negative runoff is obviously not physical and requires some interpretation from the side of the user, to make sure that there are hard boundaries.

**(6) I would appreciate some more information (figures, tables) on the actual "weights" and "biases"**

We thank the reviewer for his suggestion. We have now provided in table S1 the weights and bias ratios calculated for the participating products over a range of river basins:

*Table S1 shows examples of weights and bias ratios calculated for the participating models over a range of river basins. It shows that HBVS, JULES1, JULES2 and SURF2 didn't participate in the weighting over the large basins (i.e. Amur, Indigirka, Mississippi, Murray-Darling, Olenek, Parana, Pechora and Yenisei) since these models don't have estimates for streamflow which are needed to construct the weights over large basins. For the smaller Copper River basin, however, runoff estimates from all models participated in deriving weighted runoff estimates. Table S1 also shows that in many cases, models were assigned negative weights. While this might not be expected in typical performance-based weighting, it is possible when weighting is based on error covariance as well as their performance differences in this formulation. We show below how the weights can be modified to non-negative weights.*

*Table S1: Example of weights (w) and bias ratios (r) computed for the participating products over a range of river basins.*

| | HBVS | | HTESS1 | | HTESS2 | | JULES1 | | JULES2 | | LISF | | PCRG | | SURF2 | | SURF1 | | WGAP3 | | W3RA | |
|---|---|---|---|---|---|---|---|---|---|---|---|---|---|---|---|---|---|---|---|---|---|---|
| | w | r | w | r | w | r | w | r | w | r | w | r | w | r | w | r | w | r | w | r | w | r |
| Amur | | | -1.22 | -0.78 | 0.14 | -0.18 | | | | | 0.46 | 0.11 | 1.75 | 0.09 | | | 0.51 | -1.35 | -0.71 | -0.10 | 0.08 | -0.46 |
| Copper | 0.33 | -0.32 | -0.35 | -0.42 | 0.59 | 0.02 | -0.47 | -0.35 | -0.33 | 0.20 | 0.14 | -0.18 | 0.89 | -0.19 | 0.32 | 0.42 | 0.84 | -0.01 | 0.07 | 0.03 | -1.02 | -0.20 |
| Indigirka | | | -0.35 | -0.80 | 0.03 | 0.02 | | | | | 1.23 | -0.27 | 0.42 | -0.10 | | | 0.79 | -0.95 | -0.02 | -0.04 | -1.10 | -1.15 |
| Mississippi | | | 0.33 | -0.14 | 0.02 | -0.45 | | | | | -0.09 | 0.39 | 0.31 | 0.28 | | | -0.13 | -0.06 | 0.25 | 0.02 | 0.30 | 0.24 |
| Murray-Darling | | | 1.01 | 0.75 | 0.08 | 0.74 | | | | | -0.12 | 0.91 | 0.01 | 0.93 | | | 0.08 | 0.81 | 0.34 | 0.40 | -0.41 | 0.84 |
| Olenek | | | -0.35 | -0.76 | 0.08 | -0.12 | | | | | 2.07 | -0.16 | -0.95 | -0.18 | | | 0.28 | -0.70 | 0.10 | -0.03 | -0.22 | -0.83 |
| Parana | | | 0.26 | 0.01 | 0.16 | -0.19 | | | | | -0.88 | 0.34 | 0.18 | 0.48 | | | 0.20 | -0.12 | 1.29 | 0.06 | -0.21 | 0.36 |
| Pechora | | | -0.38 | -0.34 | 0.33 | -0.15 | | | | | 0.84 | -0.13 | 0.82 | -0.21 | | | -0.01 | -0.42 | -0.47 | -0.01 | -0.14 | -0.41 |
| Yenisei | | | -0.71 | -0.71 | -0.10 | -0.21 | | | | | 2.20 | -0.09 | -0.96 | -0.09 | | | 0.52 | -0.65 | 0.39 | -0.01 | -0.33 | -0.72 |

**(7) some of the references have artefacts, e.g. "nan" values instead of page numbers…**

**Thanks for spotting this out. We have now fixed this in the manuscript**

**Response to specific comments and questions**

**Section 2.1: The authors might also be interested in the following global-scale data source: Do et al (2018, doi:10.5194/essd-10-765-2018) & Gudmundsson et al (2018, doi: 10.5194/essd-10-787-2018). Sorry for this self-citation.**

Thanks for pointing us to this very impressive dataset. It is not immediately clear how we can use the GSIM in the current analysis since we need continuous discharge time series. However, we will work towards using the dataset in the future to improve our dissimilarity index.

**Section 2.2: What about other comprehensive model ensembles, such as ISIMIP2a (http://dx.doi.org/10.5880/PIK.2017.010)**

We agree with the reviewer that ISIMIP2a model outputs provide a suite of valuable datasets for runoff and discharge. However, we haven't included any of those in our analysis because at the beginning of our project, these datasets spanned up to 2005 only while our employed datasets spanned up to 2012.

We also note that three out of the eight models that we employed in this study are members of the ISIMIP2a models. In the future, we aim to include additional datasets, and we might consider including datasets from ISIMIP2a if we sort out the difference in the temporal coverage period (i.e. up to 2010 now for ISIMIP2a ensemble, while up to 2012 for LORA)

**page 4, line 25: Is there some empirical evidence supporting the assumption of constant bias ratio?**
Good point. The literature doesn't provide any empirical evidence that supports or contradicts this assumption. However, this assumption was a part of our whole approach that we have tested

in section (2.5). The results from Figures 4 and 5 indicate that our overall approach was succeeding.

We have now clarified this point in section 2.3:

> *We note that there is no empirical evidence in the literature that the assumptions presented in Eq 1 and Eq 2 are valid or invalid. However, they are a core part of our overall approach which we tested and demonstrated to be successful later in this paper.*

**Generall for methods: I found the mixed use of Ragg_k^j and q_k^j confusing. Would it be possible to clarify the difference/commonality once and then only use one of them?**

Over a given basin and for a participating model $k$ Ragg$_k$ refers to the spatial aggregate of simulated runoff across a basin, while q$_k$ refers to modelled streamflow at a grid cell underlying a stream gauge.
We have now clarified these terms in the text.

> *where $j \in [1, J]$ are the time steps and $k \in [1, K]$ represent the participating models, $x_k^j$ (i.e., integrated runoff $Ragg_k^j$ over the basin areas in small basins and modelled streamflow at a gauge location $q_k^j$ in large basins) is the value of the participating dataset in $m^3 \ s^{-1}$ at the $j^{th}$ time step of the $k^{th}$ participating model.*

**page 5, line 2: not really clear what you mean with this sentence. what is the number of records?**

The number of records refers to the total number of available monthly observations available for a basin. We have now clarified this in the text

> *To avoid over-fitting when applying the weighting approach, we limited the number of participating models so that the ratio of number of records (i.e. total number of available monthly observations within the period of study) to number of models does not fall below ten.*

**page 5, line 4: An alternative approach for dealing with negative values might be to log transform runoff/stream flow before doing the computations and back transform it in the end. See e.g. Gudmundsson & Seneviratne (2015, doi: 10.5194/hess-19-2859-2015; 2016, doi:10.5194/essd-8-279-2016). Again, my apologies for self-citation.**

Thanks for sharing this study. This is something to look at in the future versions of LORA and requires testing its applicability with our methods for deriving the uncertainty estimates.

**page 5, line 10: Some more details on the transformation process would be appreciated.**

**I know it is published in Hobeichi et al. (2018) but it would make the paper easier to understand if it would be outlined in more detail here.**

We thank the reviewer for his comments, we have addressed this earlier in this document. Please see our response to (5).

**page 5, line 20: transfer of "weights" from the 3 most similar basins; why 3 and not a larger/smaller number? Any empirical motivation for this? Would optimizing this hyper-parameter help to get even better results?**

We agree this is a subjective choice. The dissimilarity technique has been previously applied to find 10 donors for 1 receptor. Given that all the selected donors must have very close similarity indices, we found by trial and error that increasing the number of donor basins might introduce donor basins that have a significantly different similarity index, and that setting the number of donor basins to three seemed most appropriate. Informal optimisation. We have now clarified this in the text:

> *The dissimilarity technique has been previously applied to find ten donors for one receptor. Given that all the selected donors must have very close similarity indices, we found by trial and error that increasing the number of donor basins might introduce donor basins that have a significantly different similarity index, and that setting the number of donor basins to three seemed most appropriate. Informal optimisation.*

**page 6. line 8: Note that "Tundra" and "Subarctic" will not always have permanent ice!**
Thanks for spotting this, we have now changed the text to read:

> *areas covered with ice during most of the year (defined by climate zones Tundra, Subarctic and Ice cap)*

**Also: The chosen climate zone map is very uncommon, and most readers will not be familiar with it. Therefore, it needs to be presented in a figure. Alternative: why not use a common climate-zone definition (e.g. Köppen-Geiger?)**
We used this particular climate map because it comprises only 12 broad climate classes (compared to more than 30 in other climate maps e.g. Köppen–Geiger). This reduced the divisions made to large heterogenous basins, while ensuring that the resultant basin zones within individual basins have very distinct climate characteristics.

We have now clarified this in the text and provided a figure of this climate map in the supplemental material

> *We used this particular climate map because it comprises only 12 broad climate groups (compared to more than 30 in other climate maps e.g. Köppen–Geiger). This reduced the divisions made to large heterogenous basins, while ensuring that the resultant basin zones of individual basins have very distinct climate characteristics.*

[Figure]

**Figure S1: Climate map used in this study (available from ArcGIS online). It is a simplified climate zones map consisting of 12 broad climate classes**

**page 7. l 15: should be fig 4 not fig 3**
Thanks for spotting this. We made the changes in the text

**Fig 4: I found the "relative" improvement difficult to grasp upon first reading. Suggestion: just show the performance for WPin WPout (and omit individual models, as they are shown later)**
We agree with the reviewer that the plot in Figure 4 needs careful examination, and that one might think that Figure 4 and Figure 5 (now Figure 6) provide redundant information. However, while both Figures look similar, each achieves a different purpose. For instance, the plot in Figure 4 provides evidence that our approach (i.e. using transferred weights from the most similar gauged basins to derive runoff estimates at the receptor basins) is succeeding by the fact that $WP_{out}$ offers improvement over the individual products. On the other hand, Figure 5 (now Figure 6) compares the performance of the individual product with that of $WP_{in}$ (i.e. partially LORA) over the gauged basins. We therefore believe that it is worth providing both plots to the reader.

**page 9, line 18:**
**(1) "mean runoff uncertainty". What was exactly calculated? Note that simply averaging sqrt(sigma) will yield wrong results. Instead the rules for propagation of uncertainty should be used (https://en.wikipedia.org/wiki/Propagation_of_uncertainty). If the authors did already consider this, I apologize for this comment.**

**(2) Note that there are strong perceptions on what "reliability" means in ensemble forecasting.**
**See e.g. "reliability diagrams" and "reliability histograms"**
**page 9, line 31 & Fig 8: how did you compute the mean uncertainty? (refer to my comment above)**

Great point. We understand that averaging sqrt(sigma) cannot be used to derive the uncertainty of runoff fields from the uncertainty of streamflow or vice versa, but rather there are various rules for propagation of uncertainty that can be applied. In our response to (5) We have now explained our method for deriving uncertainty estimates for both runoff fields and their spatial aggregate (streamflow) which had not been clearly detailed in the manuscript. Here "mean runoff uncertainty" refers to the seasonal mean runoff uncertainty (climatology), we agree with the reviewer that this should be specified to avoid confusion with the spatial mean of uncertainty which does not really make sense. We have now clarified this in the text:

> *We calculated the seasonal relative uncertainty expressed as the ratio of the seasonal average uncertainty to seasonal mean runoff (i.e.* $\frac{mean\ runoff\ uncertainty}{mean\ runoff}$*) over the period 1980 – 2012.*

**page 10, line 31: not Fig 3...**

Thanks for spotting this out, we have now fixed this in the text.

---

## Author Comment (AC2) · 28 Nov 2018

**Manuscript hess-2018-386 entitled "Linear Optimal Runoff Aggregate (LORA): A global gridded synthesis runoff product"**

We would like to thank the anonymous reviewer for their constructive comments on our manuscript. This document outlines our responses to their comments and the improvements made to the manuscript.

**Response to overall comments and questions**

**(1) I miss a few explicit examples explaining why runoff is at all useful, especially at 0.5 degrees. The argument is that accurate estimates of runoff are critical to inform climate change adaptation strategies, to guide appropriate water management in agriculture and to enable the assessment of the impact of anthropogenic activities on ecosystems. However, what does runoff at 50x50 km resolution even mean? It is in terms of scale too far off from being operationally relevant. Thus, a stronger justification using examples is called for.**

We thank the reviewer for their suggestion: We have replaced the text with:

> *Characterizing its dynamics and magnitudes is a major research aim of hydrology and hydrometeorology and a critical importance to improve our understanding of the current conditions of the large-scale water cycle and predict its future states. More accurate estimates also provide additional constraint for climate model evaluation.*

**(2) I also miss some references to the earliest work on runoff fields, e.g. Fekete et al. 2002: https://agupubs.onlinelibrary.wiley.com/doi/full/10.1029/1999GB001254**

We agree with the reviewer, the study of Fekete et al. (2002) is an important example of how streamflow observation and model outputs can be combined to generate runoff fields. We have now referred to their study in the text

> *..., several other studies attempted to correct the runoff outputs directly rather than the model parameters, for example by bias-correcting model runoff outputs based on streamflow observations (Fekete et al., 2002; Ye et al., 2014),*

**(3) Regarding to previous work and scales, I would like to call the authors' attention to a recent publication by Barbarossa et al.( https://www.nature.com/articles/sdata201852). They provide discharge estimates at 1 km resolution. I move that these are probably more representative for local runoff than those obtained from GHMs at half degree resolution?**

We thank the reviewer for pointing out this study. We of course agree that FLO1k better represents small streams due to its higher spatial resolution. However, FLO1k does not necessarily provide more accurate estimates for large rivers. Additionally, FLO1k only provides information about the mean, minimum, and maximum annual flow, which limits its usefulness. In contrast, LORA provides valuable information about flow timing and the seasonal runoff distribution. The two datasets are thus quite different and in some ways complementary. In the revised paper we now cite Barbarossa et al. (2018).

**(4) Section 2.3: I find that there is too little info on the method used to compute the weights. I don't think that it should be necessary read another paper to comprehend the essentials of the methods used. So, I would want some more explanation on how the weights are calculated.**

As noted in our response to Reviewer 1 and below, we have included significantly more detail about this in the revised manuscript.

**For instance:**
**- How are correlations between models accounted for?**
An error covariance matrix is calculated for the participating models. The weights are functions of this error covariance matrix. We provide more details below
**- Are the weights allowed to be negative?**
Yes, weights can be negative

**- Is the sum of the weights adding up to one?**
Yes, the weights add up to one.
**If this is the case, one has to add another equation and transform a constrained optimization to an unconstrained one using Lagrange multipliers.**
This is correct, we constrained the weights to sum up to one, and we transformed the problem of minimizing $\sum_{j=1}^{J}(\mu_q^j - Q^j)^2$ to a problem of minimizing a function that involves a Lagrange multiplier.

We have now detailed the weighting method in the text:

> *At each gauged basin, we built a linear combination $\mu_q$ of the participating modelled streamflow datasets $x$ (i.e. Ragg in small basins and modelled streamflow, q, in large basins) that minimized the mean square difference with the observed streamflow Q at that basin such that: $\mu_q^j = \sum_{k=1}^{K} w_k(x_k^j - b_k)$ where $j \in [1, J]$ are the time steps and $k \in [1, K]$ represent the participating models, $x_k^j$ (i.e., integrated runoff $Ragg_k^j$ over the basin areas in small basins and modelled streamflow at a gauge location $q_k^j$ in large basins) is the value of the participating dataset in m3 s-1 at the $j^{th}$ time step of the kth participating model, the bias term $b_k$ is the mean error of $x_k$ in $m^3$ $s^{-1}$. The set of weights $w_k$ provides an analytical solution to the minimization of $\sum_{j=1}^{J}(\mu_q^j - Q^j)^2$ subject to the constraint that $\sum_{k=1}^{K} w_k = 1$, where $Q^j$ is the observed streamflow at the $j^{th}$ time step.*

*This minimization problem can be solved using the method of Lagrange multipliers by finding a minima for*

$$F(w, \lambda) = \frac{1}{2}\left[\frac{1}{(J-1)}\sum_{j=1}^{J}(\mu_q^j - Q^j)^2\right] - \lambda\left(\left(\sum_{k=1}^{K} w_k\right) - 1\right).$$

*The solution to the minimization of $F(w, \lambda)$ can be expressed as* $= \frac{A^{-1}1}{1^T A^{-1}1}$, *where* $1^T = $

$\overbrace{[1,1,\ldots,1]}^{k\ elements}$ *and A is the $k \times k$ error covariance matrix of the participating datasets (after bias correction), i.e.* $A = \begin{pmatrix} c_{1,1} & \cdots & c_{1,k} \\ \vdots & \ddots & \vdots \\ c_{k,1} & \cdots & c_{k,k} \end{pmatrix}$. *A is symmetric and the term $c_{a,b}$ is the covariance of the $a^{th}$ and $b^{th}$ bias corrected dataset after subtracting the observed dataset, while each diagonal term $c_{k,k}$is the error variance of dataset k. We note here that the solution presented here is based on the performance of the participating products (diagonal terms of A) and the dependence of their errors (accounted for by the non-diagonal terms of A). For derivation see Bishop and Abramowitz (2013).*

*We then derived the weighted runoff dataset by applying the computed weights on the bias corrected runoff estimates of the participating models. The weighted runoff dataset is expressed as:*

$$\mu_r^j = \sum_{k=1}^{K} w_k(r_k^j - b'_k)$$

*Where $r_k^j$ is the value of runoff estimate in $kg\ m^{-2}s^{-1}$ of the $k^{th}$ participating model at the $j^{th}$ time step and $b'_k$ is its runoff bias in $kg\ m^{-2}s^{-1}$.*

**Line 8-10 page 5: This seems to assume that travel times are less than a month because it neglects routing?**
Yes, this is worth mentioning. It is a limitation that has possibly led to an overestimation in the computed uncertainties over large basins. We have now added this to the text.

> *Given that there are no direct observations for runoff, uncertainties were computed from the discrepancy between the modelled runoff aggregates and observed streamflow. This ignored the lag time between LORA integrated runoff and observed streamflow at the mouth of the river and induced biases that possibly led to overestimated uncertainty over large gauged basins.*

**(6) Lines 11-13 page 5: "It provides better estimates than simply calculating the standard deviation of the involved products". Is that really the case? If your weighting method is optimal and you have bias-corrected correctly would the following estimator for each pixel not be unbiased (i and j are different products bias corrected):**

$$\hat{\sigma}_r^2 = \frac{1}{N \times 11 \times 11}\sum_{i=1}^{11}\sum_{j=1}^{11}\sum_{t=1}^{T} w_i w_j (r_{i,t} - \mu_r)(r_{j,t} - \mu_r)$$

**By moving a window T over time you get your time varying variance**

We thank the reviewer for his suggestion. We think that while the suggested formula provides time varying uncertainty estimates associated to the weighted runoff, it does not account for the dependence between $r_{i,t}$ and $r_{j,t}$ which is likely to lead to an overestimation of uncertainty. Meanwhile, the ensemble dependence transformation process that we applied in this paper to the participating products transforms the dependent estimates to statistically independent estimates.

We don't really have observations for runoff, so we can't test our method for deriving uncertainty on runoff, however we can test it on streamflow (i.e. runoff aggregates). We have performed out-of-sample tests to show that the distribution of the errors over the gauged basin is similar to the distribution of their errors when they are considered ungauged. We have now explained how we have performed this test and showed the results in the manuscript:

> *The uncertainty estimates computed at the gauged basins represent the deviation of (the spatial aggregate of) our weighted product ($Ragg_\mu$) from the observed streamflow, since the in-sample uncertainty estimates are calculated from the variance of the transformed ensemble, which by design equals MSE of $Ragg_\mu$ against observations (i.e. error variance of $Ragg_\mu$). To test if the uncertainty estimates perform well out-of-sample (i.e. at the ungauged basins), we took a gauged basin, but instead of constraining the weighting using observed streamflow from this basin, we constructed model weights by using the three most similar donor basins. We could then calculate MSE of $Ragg_\mu$ against observations from the three donor basins, denoted by $MSE_{in}$, which provides us with the uncertainty estimates calculated in-sample ($\sqrt{MSE_{in}}$), since the observational data used in this case is the same dataset that was used to train the weighting. We also calculated the MSE of the aggregated weighted product against the actual observation of the gauged basin and denoted this $MSE_{out}$. $\sqrt{MSE_{out}}$ represents the uncertainty estimates computed out-of-sample, since the comparison was performed against observational data that has not been used to train the weighting. We repeated the out-of-sample test for all the gauged basins.*

> *We displayed the results of the out-sample-test by showing the ratio $\sqrt{MSE_{out}}/\sqrt{MSE_{in}}$. If the approach is succeeding, we expect that this ratio is around one, indicating that the values of $MSE_{in}$ and $MSE_{out}$ are close to each other. We used a box and whisker plot, where each sample is a different basin, to show the results.*

We have also commented on the results:

> Critically though, the fact that the weighting delivers improvement over all models when the weights are transferred from similar basins indicate that the dissimilarity technique is succeeding and can be effectively used at the ungauged basins by feeding the weighting with data from the most similar basins with streamflow observations. *Furthermore, the boxplot in Fig 5 shows that, overall, when the uncertainty estimates are computed out-of-*

*sample they are very similar to what they would have been if they were computed in-sample. This demonstrates that the dissimilarity technique can be effectively used to derive not only the weighting product but also its associated uncertainties at the ungauged basin.*

[Figure]

*Figure 5: Box and whisker plots displaying the ratio of (1) the uncertainties of the spatial aggregate of the weighted product computed out-of-sample to (2) the uncertainties of the spatial aggregate of the weighted computed in-sample.*

**(7) Line 20-25 page 5: transferring the weights from donors to receptors. Are one set of weights obtained jointly for the three donor catchments or are three sets of weights averaged and transferred?**

Yes, this was not clear in the text. One set of weights is obtained jointly from the three donor catchments. We clarified this in the text.

> *We then implemented the weighting technique on the ensemble of 11 (in small basins) or eight (in large basins) model outputs by matching Ragg calculated across the selected donor basins with the observed streamflow. This resulted in one set of weights and bias ratios obtained jointly from the three donor basins.*

**(8) Line 11 on page 10. Why compare only with VIC? Why not GLDAS (4 models).**
Good question. At the time of analysis, GLDAS version1 model outputs had either a very short common period with LORA or a coarse resolution (1°) and showed a significant disagreement with observation when we interpolated them to a 0.5° grid. We clarified this in the text.

> *Other global estimates of total runoff are also available such as GLDAS and Multi-scale Synthesis and Terrestrial Model Intercomparison Project (MsTMIP; Huntzinger et al., 2016), however we haven't compared LORA with these datasets because they either have*

*a short common period with LORA, or a coarser resolution (i.e. 1o) and showed a significant disagreement with observation when interpolated to a 0.5° grid.*

**(9). Table 2: why not add the estimated total runoff volumes from GRDC (also globally in km³). It would be good to see what the global runoff volume is in this product compared to other estimates.**
Good idea. We have now added the average total yearly volume of discharged water from LORA and observation

*Table 2: A comparison of mean annual runoff (mm/year) of 16 major basins covering different climate zones around the world for LORA and VIC (Zhang et al., 2018), the yearly volume of LORA runoff aggregates (i.e. flow in Km³) and observed annual flow (Km³) over the basins and mean annual uncertainty values associated with LORA runoff are shown and the adjusted VIC annual runoff values within 5% error bounds for water budget closure are displayed. Observed annual flow is given only if data from all contributing stations is available over a whole year over for at least 17 years out of 33 years covered in this study.*

| Basin | VIC mm/year | VIC adjusted for water budget closure mm/year | LORA (Runoff) mm/year | LORA (uncertainty) mm/year | LORA yearly flow ± uncertainty Km³ | Observed yearly flow Km³ | Dominant climate |
|---|---|---|---|---|---|---|---|
| Amazon | 1048 | 1029 | 1151 | 360 | 6763 ± 2115 | - | Tropical wet |
| Amur | 135 | 129 | 219 | 115 | 428 ± 225 | 325 | Humid continental and semi-arid |
| Columbia | 318 | 293 | 333 | 101 | 218 ± 66 | 209 | Semi-arid and highlands |
| Congo | 407 | 404 | 358 | 147 | 1292 ± 532 | 1240 | Tropical wet and tropical dry |
| Danube | 272 | 265 | 260 | 125 | 199 ± 95 | 205 | Marine Humid, continental and humid subtropical |
| Indigirka | 132 | 120 | 228 | 171 | 78 ± 59 | 53 | Subarctic |
| Lena | 142 | 134 | 301 | 137 | 731 ± 332 | 557 | Subarctic |
| Mackenzie | 189 | 173 | 191 | 110 | 323 ± 186 | 294 | Subarctic |
| Mississippi | 220 | 215 | 212 | 123 | 616 ± 359 | 581 | Humid continental and humid subtropical |
| Murray-Darling | 42 | 41 | 15 | 6 | 12 ± 5 | - | Arid and semi-arid |
| Niger | 198 | 194 | 106 | 41 | 239 ± 87 | 170 | Arid, semi-arid |

| | | | | | | and tropical dry |
|---|---|---|---|---|---|---|
| Olenek | 114 | 106 | 230 | 208 | 48 ± 43 | 40 | Subarctic |
| Parana | 278 | 279 | 189 | 97 | 471 ± 247 | 600 | Marine and humid subtropical |
| Pechora | 342 | 308 | 420 | 420 | 131 ± 131 | 153 | Tundra and subarctic |
| Yenisei | 217 | 195 | 324 | 203 | 828 ± 520 | 612 | Subarctic |
| Yukon | 149 | 139 | 229 | 102 | 188 ± 83 | 214 | Subarctic |

**(10) Line 27 on page 10: reduced performance in dry climates. Apart from the reasons mentioned, another possible cause could be the fact that GHMs are probably less proficient in representing runoff processes in arid basins where often runoff is local and will not always be turned into streamflow**

Great point. We have now added this in the text to read:

> *It follows from Fig. 8 that the runoff values computed over dry climates tend to be less reliable than those in other regimes. This is perhaps due to biases in the WFDEI precipitation forcing that are propagated and intensified in the simulated runoff (Beck et al., 2017a). Another possible reason is the reduced proficiency of models in representing runoff dynamics in arid climates where runoff tends to be highly non-linearly related to rainfall and often evaporates locally without reaching a river system (Ye et al., 1997).*

---

## Author Response (AR2)

**Manuscript hess-2018-386 entitled "Linear Optimal Runoff Aggregate (LORA): A global gridded synthesis runoff product"**

We would like to thank the editor and referees for their constructive comments on our manuscript. As advised by the editor and Lukas Gudmundsson, we have made the following changes to the manuscript and supplements:

1) Increased the resolutions of the figures from 72-100 dpi to 600-1200 dpi.

2) Commented on Fig5:
   *Furthermore, the boxplot in Fig. 5 shows that, overall, when the uncertainty estimates are computed out-of-sample they are very similar to what they would have been if they were computed in-sample.* ==*Note however that the spread of results is large and that in 25% of the cases, uncertainty estimates are less than half of the in-sample results*==.

3) Added a plot for runoff uncertainty to the supplements

[Figure]

Runoff uncertainty (mm) in May 2003

| | | |
|---|---|---|
| 0-1 | 5-10 | 25-50 | 75-100 | 150-300 |
| 1-5 | 10-25 | 50-75 | 100-150 | >300 |

**Fig. S8: Global map of LORA uncertainty fields (mm) in May 2003**